# Arbovirus-vector protein interactomics identifies Loquacious as a co-factor for dengue virus replication in *Aedes* mosquitoes

**Benoit Besson** [1], **Oscar M. Lezcano**[1], **Gijs J. Overheul**[1], **Kirsten Janssen**[1], **Cornelia G. Spruijt**[2], **Michiel Vermeulen**[2], **Jieqiong Qu**[1], **Ronald P. van Rij** [1]*

**1** Department of Medical Microbiology, Radboud Institute for Molecular Life Sciences, Radboud University Medical Center, Nijmegen, The Netherlands, **2** Department of Molecular Biology, Faculty of Science, Radboud Institute for Molecular Life Sciences, Oncode Institute, Radboud University Nijmegen, Nijmegen, The Netherlands

* ronald.vanrij@radboudumc.nl

**Data Availability Statement:** Mass spectrometry proteomics data have been deposited to the ProteomeXchange Consortium via the PRIDE

## Abstract

Efficient virus replication in *Aedes* vector mosquitoes is essential for the transmission of arboviral diseases such as dengue virus (DENV) in human populations. Like in vertebrates, virus-host protein-protein interactions are essential for viral replication and immune evasion in the mosquito vector. Here, 79 mosquito host proteins interacting with DENV non-structural proteins NS1 and NS5 were identified by label-free mass spectrometry, followed by a functional screening. We confirmed interactions with host factors previously observed in mammals, such as the oligosaccharyltransferase complex, and we identified protein-protein interactions that seem to be specific for mosquitoes. Among the interactors, the double-stranded RNA (dsRNA) binding protein Loquacious (Loqs), an RNA interference (RNAi) cofactor, was found to be essential for efficient replication of DENV and Zika virus (ZIKV) in mosquito cells. Loqs did not affect viral RNA stability or translation of a DENV replicon and its proviral activity was independent of its RNAi regulatory activity. Interestingly, Loqs colocalized with DENV dsRNA replication intermediates in infected cells and directly interacted with high affinity with DENV RNA in the 3' untranslated region *in vitro* ($K_D$ = 48–62 nM). Our study provides an interactome for DENV NS1 and NS5 and identifies Loqs as a key proviral host factor in mosquitoes. We propose that DENV hijacks a factor of the RNAi mechanism for replication of its own RNA.

## Author summary

Dengue virus is a mosquito-transmitted virus endemic to the tropics and subtropics, affecting an estimated 390 million people yearly. While the mechanisms of infection, pathogenesis and immune evasion have been extensively studied in humans, replication in *Aedes* mosquitoes has received much less attention, despite being a critical step in the arbovirus transmission cycle. Here, we used a proteomic approach to identify *Aedes* mosquito proteins recruited by dengue virus non-structural proteins NS1 and NS5. In

partner repository with the dataset identifiers PXD031112 and PXD035419. Other relevant data are within the manuscript and its Supporting information files.

**Funding:** This study was financially supported by a VICI grant from the Dutch Research Council (NWO; www.nwo.nl; grant number 016.VICI.170.090 to RPvR) and by the Off-Road program of ZonMW (www.zonmw.nl; grant number 04510011910045 to BB). The funders had no role in study design, data collection and analysis, decision to publish, or preparation of the manuscript.

**Competing interests:** The authors have declared that no competing interests exist.

addition to previously established host proteins that interact with DENV in mammals, we identified Loquacious, a double-stranded RNA binding protein involved in the RNAi-based antiviral immune response of mosquitoes. Unexpectedly, our data showed that Loquacious functions as a proviral factor that is recruited to replication organelles to facilitate viral replication. We propose that DENV exploits host immune components, such as Loquacious, for its own benefit.

## Introduction

Mosquito-borne flaviviruses such as dengue virus (DENV) and Zika virus (ZIKV) are transmitted between humans by *Aedes* mosquitoes, causing major public health and economic burden in the tropics and subtropics [1]. Whereas *Aedes aegypti* is considered the most common vector for DENV transmission, vector competence of *Aedes albopictus* and its ability to adapt to colder climates raise concerns of a shift of arbovirus endemic regions toward the Northern hemisphere, further enhanced by global warming [2–5]. As no specific treatment against DENV is available and vaccine development faces many obstacles [6], there is a need to understand the mechanisms of flavivirus replication in mosquitoes to develop transmission blocking strategies.

Flaviviruses have a single-stranded (ss), positive-sense RNA genome that is translated upon release into the host cell and replicated at the membrane of the endoplasmic reticulum (ER), where viral proteins hijack the host cell machinery to form membrane invaginations functioning as viral replication organelles [7]. While structural proteins are essential for viral entry, assembly and release of viral particles, non-structural (NS) proteins together with host factors are required for viral RNA replication and modulation of host cell functions. Expressed in the lumen of the ER and released outside of the host cell, NS1 is an important pathogenicity factor in mammals [8,9]. In addition, NS1 contributes to the formation of viral replication organelles and recruits the oligosaccharyltransferase (OST) complex, as well as mRNA translation and protein folding factors [10–12]. As the viral RNA-dependent RNA polymerase, NS5 cooperates with other viral and host proteins to replicate the viral RNA, but also interacts with host proteins to suppress Jak-STAT signaling and modulate the spliceosome [13].

For transmission to a mammalian host, arboviruses need to efficiently replicate in their mosquito vector, which, like in mammals, requires extensive virus-host protein-protein interactions. Flavivirus host factors have been extensively studied in mammalian models using proteomics [10,13–15] as well as other high-throughput approaches such as RNAi and CRISPR screens [11,16–19]. In comparison, few studies have addressed flavivirus host factors in mosquitoes. While it is likely that flaviviruses use homologous host proteins and cellular processes in mosquitoes and mammals, this has only been confirmed for few proteins, such as SEC61 [14,19]. Given the evolutionary distance, the different physiologies and diverse immune systems of mosquitoes and mammals, it is to be expected that flavivirus proteins additionally interact with specific sets of proteins in mosquito vector and vertebrate host.

A key element differentiating mosquito and mammalian immunity is the crucial role of RNAi in antiviral defense in insects [20–22]. Double-stranded (ds)RNA formed during viral replication is cleaved by the nuclease Dicer-2 into small interfering (si)RNA duplexes that are loaded into Argonaute 2 (Ago2) to guide the recognition and cleavage of complementary viral RNA. Loading of siRNA duplexes into Argonaute proteins is facilitated by the paralogous dsRNA-binding proteins R2D2, Loquacious (Loqs) and Loqs2, the latter of which is unique to *Aedes* mosquitoes [23–27].

The *Ae. aegypti Loqs* gene encodes multiple, non-redundant spice isoforms [24], of which Loqs-PB along with R2D2 facilitates siRNA processing, whereas Loqs-PA is required for microRNA (miRNA) processing [24]. In line with a function in siRNA processing, it was found that silencing *R2D2*, *Dicer-2* and *Ago2* increases DENV replication [28]. In addition, the mosquito specific Loqs2 interacts with Loqs and R2D2 and is essential to control systemic flavivirus infections *in vivo* [23]. In the current manuscript, we use the *Loqs* transcript annotation of the *Ae. aegypti* reference genome AaegL5. Note that the *Loqs* annotation in AaegL5 does not correspond to the annotation used in [24]. The transcript referred to as *Loqs-RA* by Haac *et al.* [24] corresponds to *Loqs-RB* in AaegL5 and *vice versa*. The transcript referred to as *Loqs-RC* by Haac *et al.* does not exist in AaegL5.

To identify host proteins interacting with DENV proteins in mosquitoes, we purified FLAG-tagged DENV NS1 and NS5 expressed in *Aedes* cells in the context of all DENV non-structural proteins, followed by quantitative mass spectrometry. We identified Loqs as an essential co-factor for DENV and ZIKV RNA replication, independent of its function in RNAi. We showed that Loqs interacts with the 3' untranslated region of DENV RNA and localizes to viral replication organelles during infection of *Aedes* cells. Our data provide new insights into flavivirus replication in mosquitoes and illustrate a case in which a potential anti-viral protein, Loqs, is recruited by a virus for its own benefit.

## Results

### Interactome of DENV NS1 and NS5 proteins in *Aedes* mosquito cells

To identify host factors that interact with DENV proteins in an RNA-independent manner in mosquitoes, we established a system to express all non-structural proteins in *Ae. albopictus* cells. Expression of non-structural proteins in the absence of viral RNA is sufficient to induce membrane rearrangements, reminiscent of those induced during viral infections [7,29]. All non-structural genes were cloned under the control of an *Ae. aegypti* poly-ubiquitin promoter (PUb), introducing a 3xFLAG tag at the N-terminus of NS1 after the ER localization signal and conserving the signal peptidase cleavage site, or at the C-terminus of NS5, according to previous work [10,13] (Fig 1A). Expression of the recombinant proteins in RNAi-deficient *Ae. albopictus* C6/36 cells was confirmed by western blot, and both FLAG-tagged NS1 and NS5 were efficiently concentrated by FLAG immunoprecipitation (Fig 1B). As expected, NS1 and NS5 were expressed in distinct cell compartments in C6/36 cells (Fig 1C).

We characterized the interactome of FLAG-tagged NS1 and NS5 in C6/36 cells by FLAG immunoprecipitation, followed by label-free mass spectrometry (Fig 1D and 1E and S1 Table). To prevent false-positives due to perturbation of the cellular proteome by DENV non-structural proteins [30], we used untagged DENV non-structural proteins as a control. NS1 interacted with 55 proteins, including the viral NS2A, NS3, NS4A and NS4B proteins, whereas NS5 interacted with 45 proteins, including NS1, NS2A, NS3 and NS4A, and a total of 15 host proteins interacted with both NS1 and NS5 (Figs 1D, 1E and 2A and S1 Table). NS2B peptides were not detected, likely due to limitations of detection. However, given that NS2B is a cofactor for the NS3 protease and polyprotein processing was efficient (Fig 1B), NS2B is likely expressed at physiologically relevant levels. The identification of viral non-structural proteins in NS1 and NS5 immunoprecipitations confirmed the formation of macromolecular complexes in the absence of structural proteins and viral RNA in our experimental system.

Biological processes and functional protein network analysis identified several functional protein clusters in association with NS1 and NS5 (Fig 2 and S2 Table). Among the cellular interactors enriched at least two-fold in NS1 immunoprecipitations, we identified 11 proteasome subunits and five proteins of the OST complex (Figs 1D, 1E and 2 and S1 Table):

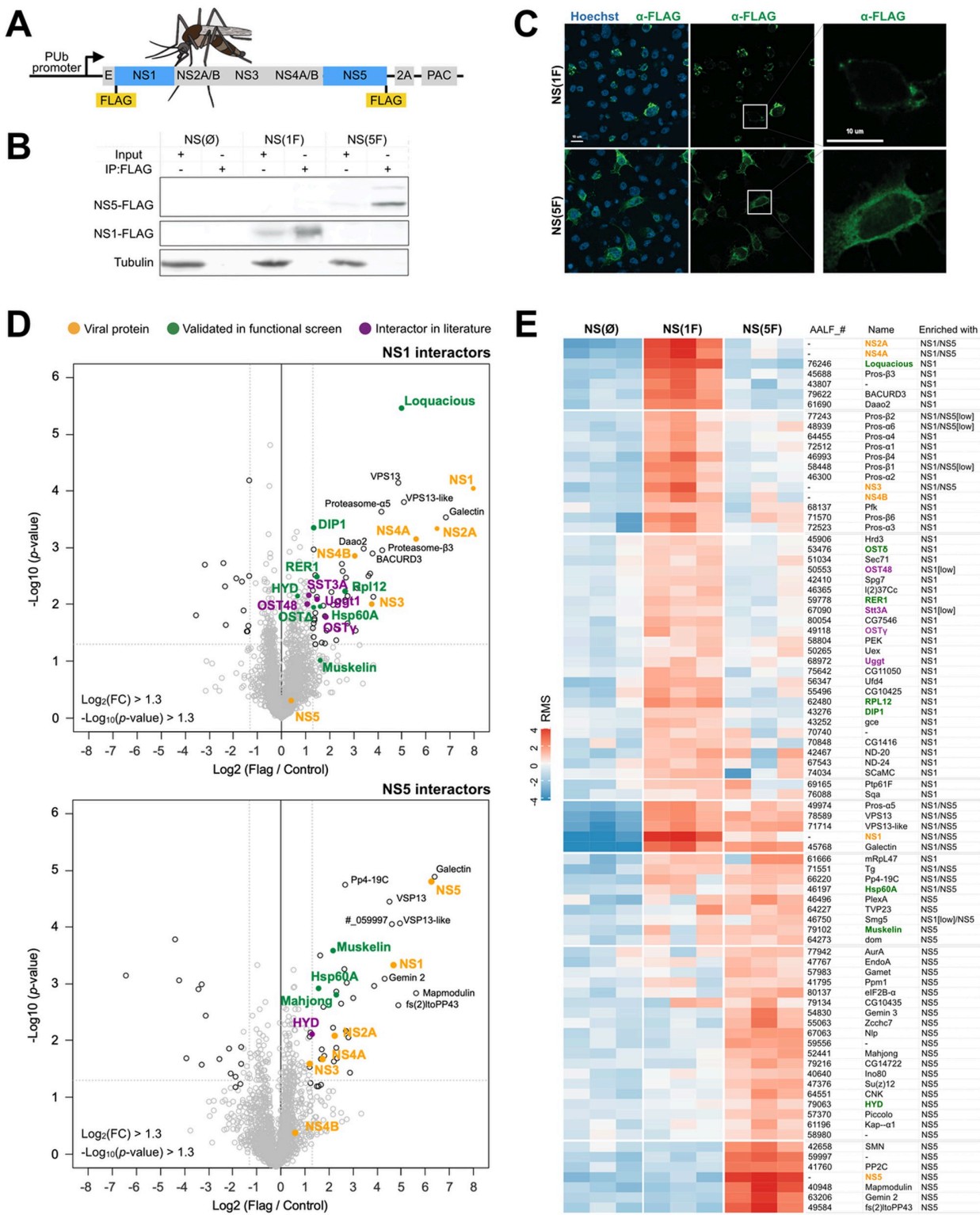

**Fig 1. Interactome of NS1 and NS5 in *Aedes* mosquito cells. A.** Schematic representation of constructs used to express DENV non-structural proteins in mosquito cells. Constructs were generated with a 3xFLAG tag at the N-terminus of NS1 (NS(1F)) or the C-terminus of NS5 (NS(5F)). A construct without a tag (NS(ØF)) was included as a control. 2A, self-cleaving peptide from foot-and-mouth disease virus; PAC, Puromycin N-acetyltransferase. Mosquito image by Mariana Ruiz Villarreal, licensed under CC0. Fig: https://bioicons.com/?query=mosquito. **B.** Western blot of input and FLAG immunoprecipitation samples of C6/36 cells expressing NS(ØF), NS(1F) and NS(5F), stained with FLAG and α-tubulin (Tub)

antibodies. See uncropped gel in S5 Fig. **C.** Confocal microscopy image of FLAG-tagged NS1 and NS5 in C6/36 cells at 24 h after transgene transfection. Cells were stained with anti-FLAG M2 antibody (green) and Hoechst to stain nuclei (blue). **D.** Volcano plot of proteins interacting with 3xFLAG-tagged NS1 (top) or NS5 (bottom) in C6/36 cell lysates as determined by label-free quantitative mass spectrometry. The X-axis shows the log2 fold change (FC) over untagged NS(ØF) (control), and the Y-axis shows -log10($p$-value). Proteins in the top right are identified as significantly enriched proteins. Colored dots indicate proteins of interest. Each condition was performed in triplicate. Proteins are named according to the fly ortholog, as defined in S1 Table. **E.** Heatmap of the relative enrichment (red) and depletion (blue) of proteins in each sample, based on row-mean subtraction and K-means clustering. Statistically significant enrichment in the volcano plot analysis is indicated, $p < 0.05$. Ost48 and Stt3A were included, although not significantly enriched. 'Low' indicates an enrichment between 2 and 2.5-fold, below the threshold of the volcano plot.

Oligosaccharide transferase Δ subunit (OstΔ, ortholog of human ribophorin II, RPN2), Oligo-saccharide transferase γ subunit (OSTγ, ortholog of human MAGT1), UDP-glucose-glycopro-tein glucosyltransferase (UGGT, ortholog of human RPN1), Oligosaccharyltransferase 48kD subunit (OST48, ortholog of human dolichyl-diphosphooligosaccharide-protein

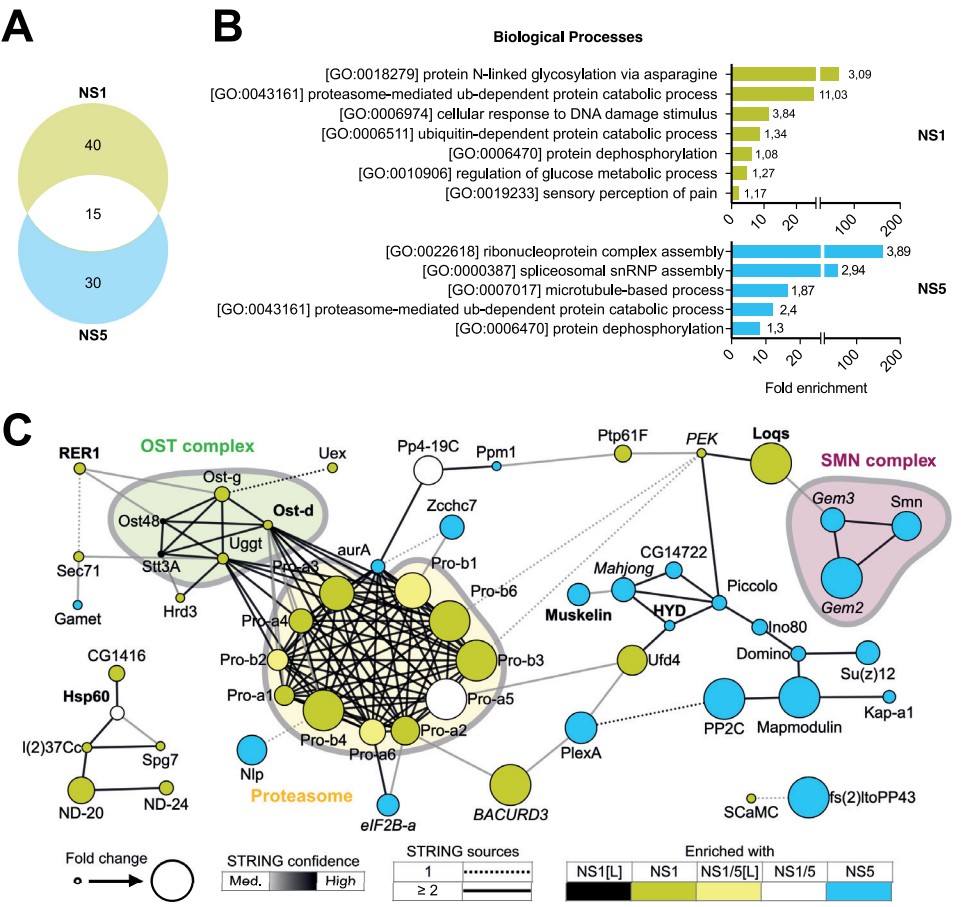

**Fig 2. Characterization of DENV NS1 and NS5 interactomes. A.** Venn diagram of 85 interactors of NS1 and/or NS5 identified by mass spectrometry (Fig 1E). **B.** GO term analyses of interactors of NS1 (top panel) and NS5 (lower panel). Enrichment of biological processes was based on *D. melanogaster* ortholog annotation. Numbers indicate -log$_{10}$ $p$ values. See complete list of GO terms in S2 Table. **C.** Functional STRING networks based on *D. melanogaster* ortholog annotation, using the following four sources: *text mining*, *experiments*, *databases* or *co-expression*. Hits were classified and colored according to their enrichment in ≥ 2 out of 3 NS(1F) or NS(5F) samples in the heatmap of Fig 1E. Node sizes represent the fold enrichment in NS(1F) or NS(5F) immunoprecipitation, keeping the highest value if the interactor was present in both. Edges are representative of the number of sources (solid or dashed) and the confidence (color) supporting the interaction as defined by STRING. Font indicates hits confirmed (bold) or not (italic) as modulators of DENV in the functional screening (Fig 3).

glycosyltransferase non-catalytic subunit, DDOST) and catalytic subunit 3A of the oligosaccharyltransferase complex (STT3B).

Amongst the cellular interactors enriched in NS5 immunoprecipitations, we identified three elements of the survival of motor neuron (SMN) complex involved in the spliceosome and small nuclear ribonucleoprotein (snRNP) assembly (Gemini 2, Gemini 3 and Smn) as well as the E3 ligase HYD (ortholog of human UBR5). In mammals, the OST complex and proteasome have previously been described as proviral host factors interacting with DENV NS1 [10,31–33], NS5 was shown to hijack the snRNP and UBR5 was purified with NS5 [13], illustrating virus-host interactions that occur both in the mosquito vector and human host. Together, these data validate our experimental model to identify physiologically relevant protein-protein interactions between virus and mosquito vector.

## Identification of DENV proviral and antiviral host factors

From the 79 host proteins interacting with NS1 and/or NS5, we selected 22 hits for a functional knockdown screen in the RNAi competent *Ae. albopictus* U4.4 cell line (Fig 3A). Cells were transfected with dsRNA, infected with DENV, and viral RNA was quantified by RT-qPCR. From the initial screen (Fig 3B), 10 hits were selected for a confirmation screen with a second set of dsRNA, targeting a different region of the gene to rule out off-target effects (Fig 3C). We classified genes as high or low-confidence hits, based on selective criteria for gene knockdown

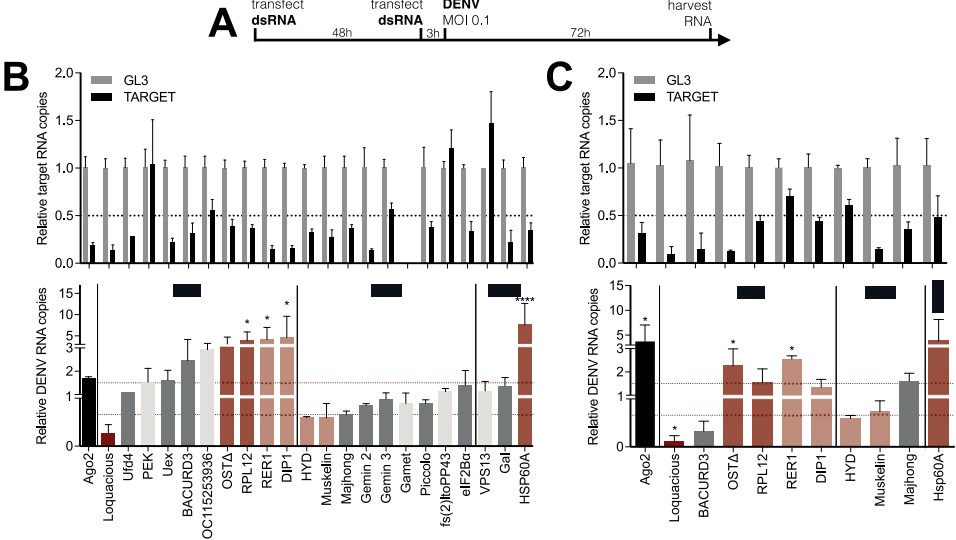

**Fig 3. Functional screen identifies Loqs as a DENV proviral host factor in *Aedes* mosquitoes. A.** Schematic outline of the functional RNAi screen. **B-C.** Relative quantification of target gene expression (top panel) and DENV RNA levels (lower panel) in U4.4 cells upon silencing of the indicated genes. Selected genes from the initial screen (B) were tested in an independent validation screen (C) using dsRNA targeting a different region of the gene. Expression was quantified by RT-qPCR, normalized to the house-keeping gene *ribosomal protein L5*, and expressed relative to expression in cells treated with dsRNA targeting firefly luciferase (GL3). *Ago2* was used as a positive control. Data represent means and standard deviation of three replicates. Color coding represents classification of hits based on gene knockdown efficiency, phenotype, and consistency between screens. Light grey, inefficient knockdown ($< 0.5$-fold); dark grey, no phenotype despite efficient knockdown ($> 0.5$-fold); dark red, strong hit with efficient knockdown and DENV RNA levels $< 0.66$-fold or $>1.5$-fold in both dsRNA sets; light red, weak hits for which one of the criteria was not met. One-way ANOVA were used to determine statistically significant differences with the GL3 control, with: * $p < 0.05$; ** $p < 0.01$; *** $p < 0.001$; **** $p < 0.0001$.

efficiency, effect of gene knockdown on DENV RNA levels, and consistency between both dsRNA data sets.

One NS1 interactor was identified as proviral factor as silencing consistently resulted in a decrease in viral RNA levels (Fig 3B and 3C), Loquacious (Loqs, ortholog of human TARBP2) as well as two NS5 interactors: the E3 ubiquitin-protein ligase HYD (ortholog of human UBR5) and Muskelin. Four NS1 interactors were found to be antiviral as silencing consistently resulted in an increase in viral RNA levels (Fig 3B and 3C): OSTΔ (RPN2), RNA polymerase I subunit H or RPL12 (ortholog of human RPA12), DISCO-interacting protein 1 (DIP1, ortholog of human ADARs), ER retention protein RER1, in addition to one common NS1/NS5 interactor, HSP60A.

Among the hits, knockdown of the RNAi cofactor *Loqs* had the strongest phenotype with up to 70–90% inhibition of DENV RNA levels (Fig 3B and 3C). Two potential Loqs host interactors were also included in the functional screen (Fig 2). *Pancreatic eIF2A-α kinase* (*PEK*), distant ortholog of human PRKRA known to interact with the human *Loqs* ortholog TARBP2, was not efficiently silenced, and knockdown of RNA helicase *Gemini 3* (ortholog of human DEAD box helicases like DDX20) [34–37] did not modulate DENV replication (Fig 3C). The finding that Loqs is a proviral host factor was unexpected, given that Loqs is an essential cofactor of RNAi, a pathway with antiviral activity in mosquitoes. For this reason and because of the strong NS1 association (Fig 1D and 1E) and consistent phenotype (Fig 3B and 3C), we focused the remainder of the study on Loqs, using *Ae. aegypti* as a model, for which a better genome annotation is available than for *Ae. albopictus*.

## Loqs is a flavivirus proviral factor in *Ae. aegypti*

The *Loqs* gene (AAEL008687) encodes three splice isoforms, which differ from each other by the presence or absence of short exons. *Loqs-RA* and *Loqs-RB* are very similar to each other, except that the 144 nt-long exon 5 is retained in *Loqs-RA*. *Loqs-RC* is a shorter isoform, in which exon 6 is retained, but exon 7 is skipped (Fig 4A). As a consequence, the gene products Loqs-PA and Loqs-PB contain two dsRNA-binding domains (dsRBD, IPR014720) as well as a Staufen, C terminal domain (IPR032478), which may be involved in protein-protein interactions. Loqs-PC contains only the two N-terminal dsRBDs.

From the mass spectrometry data, we identified one peptide unique to Loqs-PA in NS1 immunoprecipitations, whereas other peptides were shared between the three isoforms (Fig 4B). Using primers flanking exon 5 (Fig 4C and S3 Table) as well as unique primers to distinguish the isoforms (Fig 4D and S3 Table), we showed that *Loqs-RA* and *Loqs-RB* are expressed at similar levels in *Ae. aegypti* Aag2 cells, whereas *Loqs-RC* is only lowly expressed, in line with its low expression *in vivo* (S2A Fig). Together, these results indicate that NS1 interacts with Loqs-PA, although we cannot rule out interactions with Loqs-PB and Loqs-PC (Fig 4B). The reverse experiment, Loqs-PA immunoprecipitation from DENV infected Aag2 cells followed by mass spectrometry (S1 Fig and S4 Table) identified the RISC-associated proteins R2D2, Loqs2, Ago1, Dicer-1, and Dicer-2. Of interest, several RNA-binding proteins interacted with Loqs, including DIP1 found to interact with NS1 in Fig 1D, Loqs2 previously found to interact with Loqs by Olmo *et al.* [23], and AAEL004859 which was shown to have a strong antiviral effect on DENV [38]. Notably, this analysis confirmed the interaction of Loqs with DENV non-structural proteins NS1, NS3 and NS5, enriched by 4.9-, 6.1- and 10.7-fold, respectively.

To analyze which of the Loqs isoforms is responsible for the observed proviral phenotype, we compared DENV replication after RNAi-mediated knockdown of specific *Loqs* isoforms as well as other components of the miRNA and siRNA pathways in Aag2 cells (Fig 4E). Silencing of *Ago2* resulted in an increase in viral replication, in agreement with the antiviral activity of

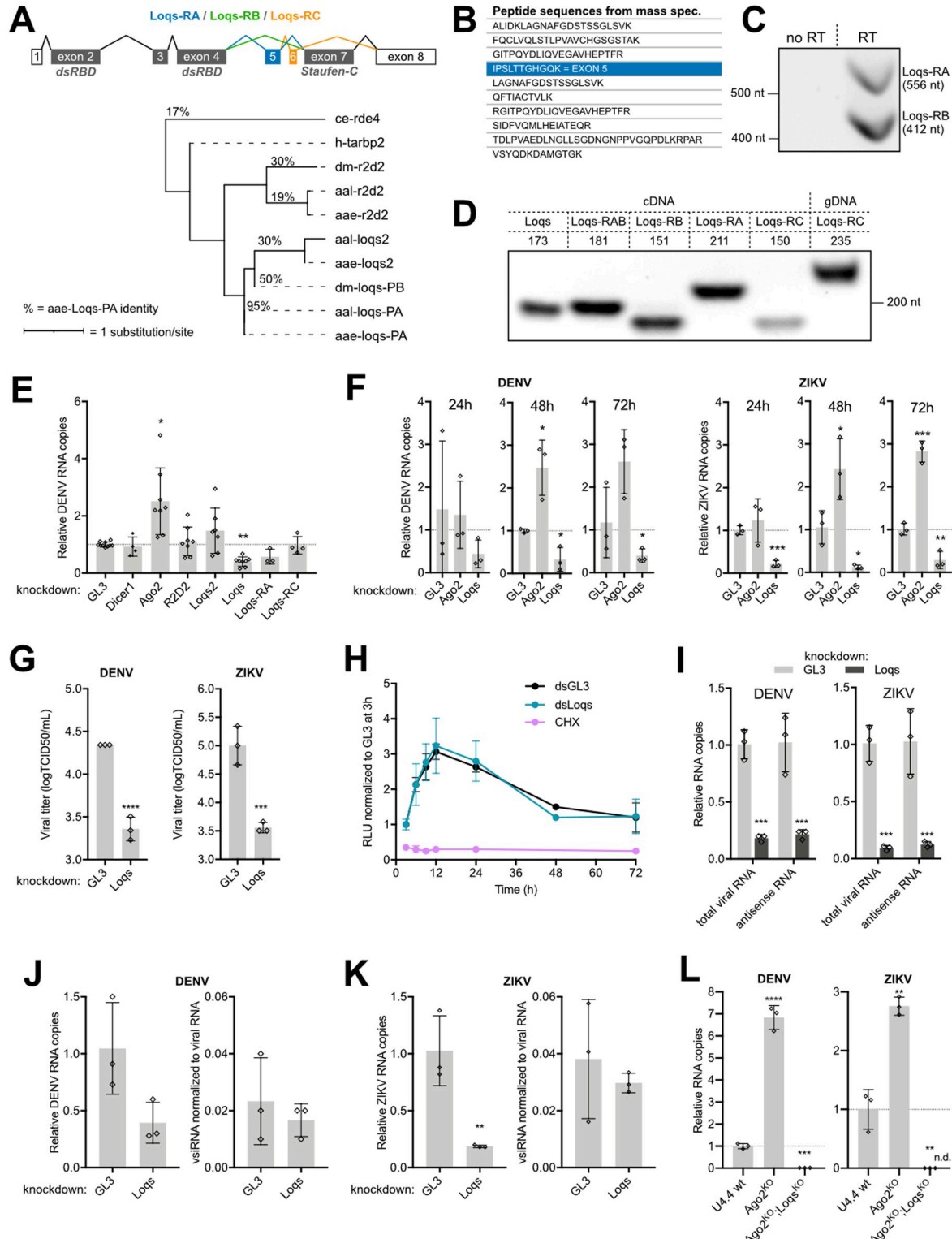

**Fig 4. Loquacious is an essential co-factor for flavivirus replication. A.** Structure of *Loquacious* splice variants in *Ae. aegypti* and maximum likelihood phylogenic tree based on the protein sequence of Loqs-PA and its orthologs and paralogs. ce, *Caenorhabditis elegans*; dm, *Drosophila melanogaster*; h, *Homo sapiens*; aae, *Ae. aegypti*; aal, *Ae. albopictus*. Percentages indicates the identity between protein sequences compared to *Ae. aegypti* Loqs-PA. Branch lengths are proportional to the number of substitutions per site. *Loqs* transcript annotation is according to the reference genome AaegL5, which differs from the annotation used in [24]. dsRBD, dsRNA-binding domain.

**B.** Loqs peptides identified by mass spectrometry in NS1 immunoprecipitations. The peptide unique to Loqs-PB is indicated in blue; other peptides are shared between *Loqs* isoforms. **C.** *Loqs-RA* and *Loqs-RB* specific amplicons from PCR using primers spanning exon 5 on cDNA from Aag2 cells. **D.** PCR amplification of *Loqs* splice variants with various set of primers on cDNA or genomic DNA isolated from Aag2 cells. Numbers indicate expected sizes. **E.** Relative quantification of DENV RNA at 72 h infection of Aag2 cells in which siRNA and miRNA pathway genes were silenced. **F.** Relative quantification of DENV RNA or ZIKV RNA at the indicated time after infection in *Ago2* or *Loqs* depleted Aag2 cells. **G.** Titration of DENV (left) and ZIKV (right) from the supernatant of *Loqs* depleted Aag2 cells after 48h of infection at an MOI of 0.01. **H.** *Renilla* luciferase activity in Aag2 cells transfected with DENV2 replicon RNA and dsRNA targeting *Loqs* (dsLoqs) or firefly luciferase (dsGL3) as a control. Luciferase activity was assessed at the indicated time points and normalized to dsGL3-treated cells at 3 h after transfection. Cells treated with cycloheximide (CHX) were included as control. **I.** Relative quantification of DENV (left) or ZIKV (right) total viral RNA or antisense viral RNA after infection of *Loqs* depleted Aag2 cells of infection at an MOI of 0.01. **J-K.** Relative quantification of DENV (H) and ZIKV (I) RNA (left) and vsiRNAs (right) at 48 h after infection of *Loqs* depleted or control (GL3 dsRNA treated) Aag2 cells. vsiRNAs were normalized to the cellular piRNA tapiR1 and viral RNA to correct for library size and viral RNA levels. **L.** Relative quantification of DENV (left) and ZIKV (right) RNA levels at 48 h after infection in wild type (wt), U4.4-*Ago2*^KO, or U4.4-*Ago2*^KO; *Loqs*^KO cells at an MOI of 0.01. ZIKV was undetectable (n.d.) in U4.4-*Ago2*^KO; *Loqs*^KO cells and relative viral RNA copies were calculated using imputed Ct values of 40. Viral RNA levels were quantified by RT-qPCR, normalized to the housekeeping gene *lysosomal aspartic protease* (E-F, I-L) and expressed relative to cells treated with dsRNA targeting luciferase (dsGL3) as negative control. See relative knockdown efficiencies in S2 Fig. Bars represent mean and standard deviation from at least three biological replicates. Non-parametric one-way ANOVA (E) and two-tailed student's t-tests (F, I-L) were used to determine statistically significant differences with the control: * $p < 0.05$; ** $p < 0.01$; *** $p < 0.001$; **** $p < 0.0001$.

the RNAi pathway, whereas silencing *Dicer-1*, responsible for miRNA maturation, and *R2D2* and *Loqs2* did not affect DENV replication. In contrast, silencing of all *Loqs* isoforms combined resulted in a 60% reduction in DENV RNA levels (Fig 4E), confirming our observations in *Ae. albopictus* U4.4 cells (Fig 3B and 3C). Furthermore, silencing of the *Loqs-RA* isoform specifically was associated with a 43% reduction in DENV RNA levels, whereas *Loqs-RC* silencing did not affect DENV replication, in line with its low expression (Fig 4E). Due to the *Loqs* gene structure, it was impossible to design dsRNA that specifically targets *Loqs-RB* but not the other isoforms. These results indicate that Loqs-PA and likely Loqs-PB are proviral host factors for DENV replication in *Ae. aegypti*.

To further characterize the role of Loqs in flavivirus replication, we compared the effect of *Loqs* silencing on DENV and ZIKV replication. *Loqs* silencing reduced viral RNA levels for both viruses at all time points analyzed (Fig 4F and S2C Fig), although the effect of *Loqs* silencing was consistently stronger for ZIKV (up to 90% inhibition at 48 h). Importantly, *Loqs* silencing resulted in a significant, > 9- and 33-fold reduction of infectious particles in the culture supernatant for DENV and ZIKV, respectively (Fig 4G and S2D Fig). These results suggest that Loqs is a proviral host factor for multiple flaviviruses in *Aedes* mosquitoes.

## Loqs does not affect RNA stability

To analyze the effect of Loqs on viral RNA stability and translation, we monitored translation efficiency of DENV RNA in cells in which *Loqs* was silenced [39]. To this end, *Renilla* luciferase expression was monitored after transfection of subgenomic replicon RNA into *Ae. aegypti* Aag2 cells, under experimental conditions in which replicon RNA is translated but not replicated (Fig 4H). Luciferase expression from direct translation of the transfected viral RNA was similar in cells treated with *Loqs* dsRNA or control dsRNA, whereas luciferase expression was blocked upon treatment with the translation inhibitor cycloheximide (CHX), as expected. These results suggest that Loqs does not affect viral RNA stability and translation efficiency, but that it is required for viral RNA replication.

To confirm the role of Loqs in DENV and ZIKV replication, we quantified the negative strand, antigenomic RNA and found that negative-strand synthesis was strongly inhibited upon *Loqs* knockdown (Fig 4I and S1D Fig). Taken together, our data suggest that Loqs is involved in the replication of viral RNA.

## Loqs proviral activity is independent of its RNAi regulatory functions

Having established the importance of Loqs for flavivirus replication, we further characterized its role during viral replication. As Loqs is a co-factor of the antiviral siRNA pathway [24], it is unlikely that its proviral phenotype is dependent on RNAi. To directly test this, we sequenced small RNAs produced in *Loqs*-depleted Aag2 cells infected with DENV or ZIKV. To account for differences in virus accumulation upon *Loqs* silencing, virus-derived siRNA (vsiRNA) levels were normalized to virus RNA levels in the same sample used for small RNA sequencing. Despite efficient silencing of *Loqs* expression (S2E Fig) and consequent reduction of viral RNA levels, no differences were observed in normalized DENV or ZIKV-derived vsiRNA levels (Fig 4J and 4K). Moreover, we did not observe major differences in miRNA levels upon *Loqs* knockdown (S2F and S2G Fig), as also observed previously when both isoforms were depleted [24].

To provide further genetic support that the proviral activity of Loqs is independent of RNAi, we used CRISPR/Cas9 gene editing on U4.4 cells lacking functional Argonaute 2 (Ago2^KO cells) to introduce frameshift mutations in *Loqs* (S3 Fig), thus generating U4.4 cells in which both *Ago2* and *Loqs* are functionally inactive (U4.4-*Ago2*^KO; *Loqs*^KO). As expected, a strong increase in viral replication was observed for both viruses in U4.4-*Ago2*^KO cells compared to wildtype (wt) U4.4 cells. In contrast, viral RNA levels were strongly reduced in U4.4-*Ago2*^KO; *Loqs*^KO cells, indicating that the proviral activity of Loqs is independent of its function in the RNAi pathway and essential for viral replication (Fig 4L).

## Loquacious colocalizes with dsRNA in viral replication organelles

Considering that Loqs contains multiple dsRBD domains, we explored the potential role of Loqs as a cofactor of viral RNA replication. We analyzed subcellular localization of Loqs-PA and Loqs-PB in mock and DENV infected Aag2 cells (Fig 5A). In non-infected cells, GFP-tagged Loqs-PA or Loqs-PB showed a discrete, punctate staining across the cytoplasm as well as lower, diffuse staining in the cytoplasm. Moderate dsRNA signal was detected in the nucleus, likely corresponding to sites of high production of structured cellular RNA, which was clearly distinct from the strong cytoplasmic dsRNA signal in infected cells. In DENV-infected cells, both Loqs isoforms showed a punctate pattern in the cytoplasm, but the signal was strongly enriched in perinuclear puncta. Importantly, for both isoforms a strong colocalization with viral dsRNA was observed, suggesting that Loqs relocalizes to viral replication organelles in infected cells (Fig 5B).

We next investigated whether Loqs directly interacts with DENV RNA. To this end, we purified recombinant Loqs-PA and Loqs-PB as a fusion protein with maltose binding protein (S4 Fig) and performed electrophoretic mobility shift assays (EMSA). As expected, Loqs-PA and Loqs-PB bound specifically and with high affinity (2–18 nM) to control dsRNA in gel mobility shift assays (Fig 6A and 6B). We next incubated Loqs with *in vitro* transcribed single-stranded RNA (ssRNA) corresponding to the DENV 5' untranslated region (UTR), the 3'UTR, as well as coding sequences in the NS1 and NS5 genes, and complexes were resolved on native polyacrylamide gels. We found that both Loqs isoforms bound with high affinity (48–62 nM) to the DENV 3'UTR, whereas binding to the other RNAs was much less efficient (Fig 6C). Interestingly, multiple Loqs-3'UTR complexes were formed at higher Loqs concentrations, suggesting that Loqs may bind to multiple RNA structures in the 3' UTR. Altogether, we propose that DENV non-structural proteins recruit Loqs to viral replication organelles, where it interacts likely through its dsRNA-binding motifs with viral RNA to facilitate viral RNA replication.

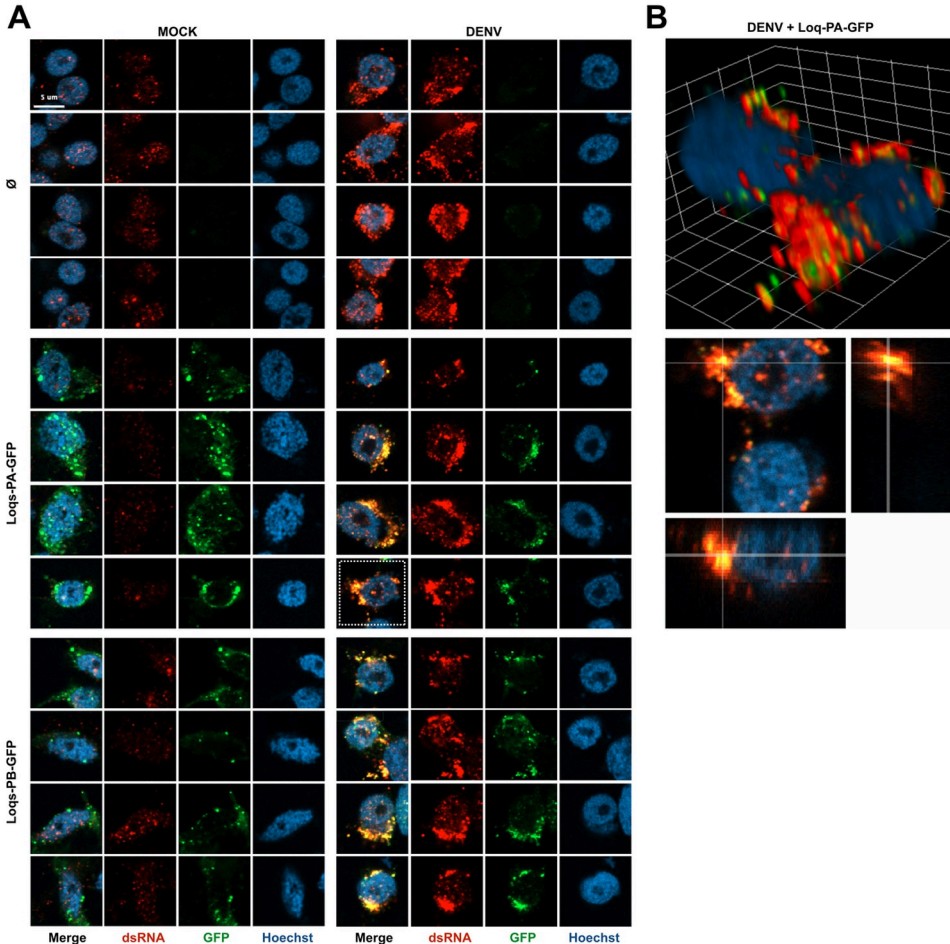

**Fig 5. Loquacious colocalizes with DENV replication organelles. A.** Confocal microscopy images of mock or DENV infected Aag2 cells. Cells were transfected at 72 h post infection with plasmids encoding the indicated transgenes or mock transfected (Ø) and processed for microscopy 24 h later. Cells were stained with anti-dsRNA J2 antibody (red) and Hoechst to stain nuclei (blue). Scale bar corresponds to 5 μm and is the same for all panels. In mock infected cells, a nuclear background signal is detectable that is distinct from the cytoplasmic viral dsRNA signal in DENV infected cells. **B.** Three-dimensional visualizations of two DENV infected Aag2 cells transfected with Loqs-PA-GFP indicated with a dashed square in panel A. The dsRNA and GFP signals were optimized for visualization in the three-dimensional projection. The grid is scaled with 2 μm xyz units.

## Discussion

Arboviruses replicate efficiently in their invertebrate vectors as well as in their vertebrate hosts, and viral RNA and proteins thus interact with cellular proteins from these evolutionary diverse hosts. In this study, we used a proteomic approach to identify proteins interacting with DENV NS1 and NS5 in *Aedes* mosquitoes. Among the interactors, we identified the dsRNA binding protein Loquacious as a DENV and ZIKV proviral factor that interacts with viral RNA at the 3' UTR. We propose that Loquacious is recruited to replication organelles to facilitate viral RNA replication. Loquacious is a cofactor of RNA silencing pathways [24] and our data thus suggest that DENV exploits proteins of an antiviral immune response for its own benefit. The human Loqs ortholog TARBP2 (also known as TRBP), a cofactor for HIV-1 replication that binds to the structured TAR RNA element [40], has been found to

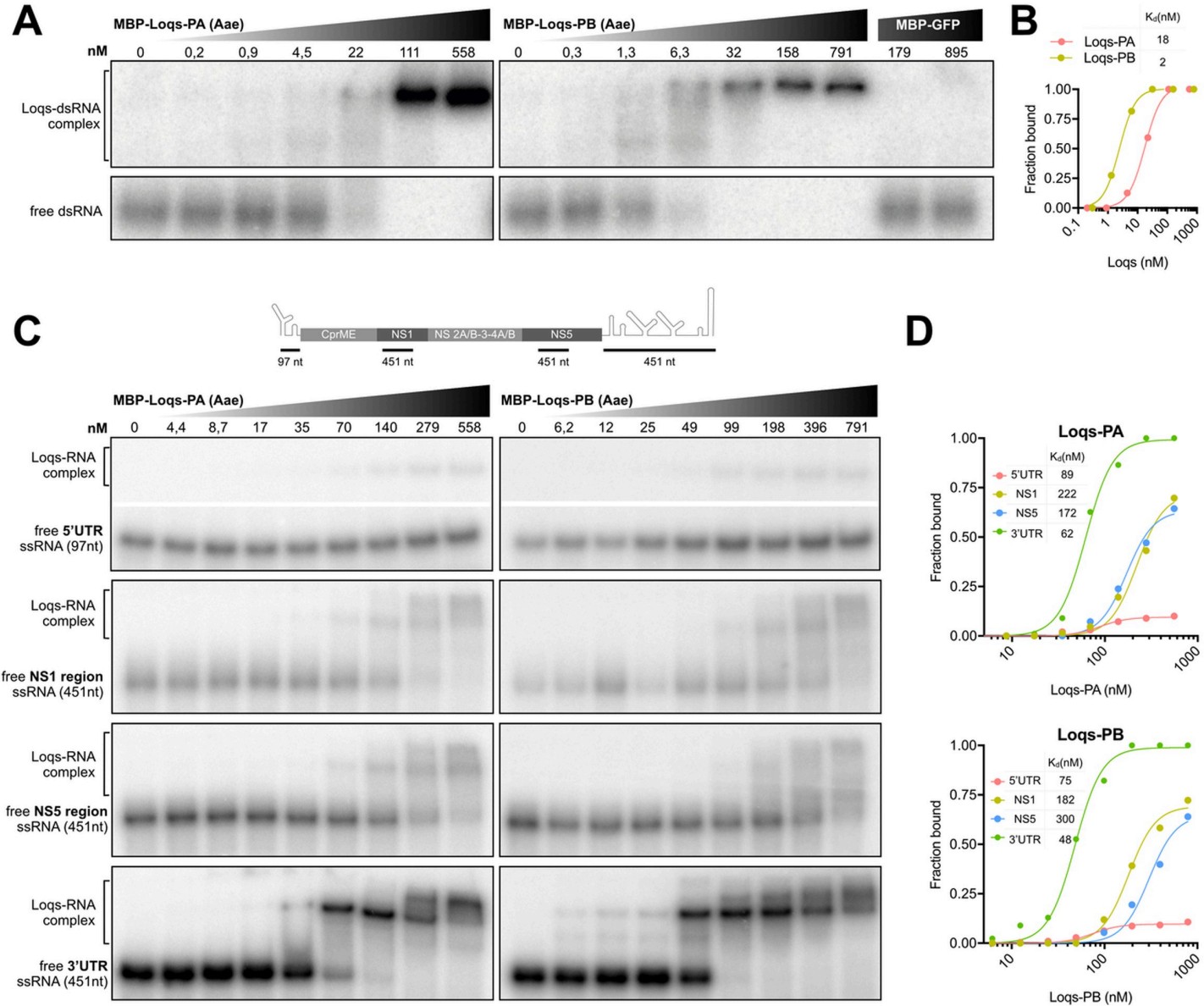

**Fig 6. Loquacious interacts directly with DENV 3'UTR. A.** Electrophoretic mobility assay of *Ae. aegypti* Loqs-PA (left panel), Loqs-PB (right panel) and GFP control with a 117 bp [32]P labeled dsRNA corresponding to the firefly luciferase sequence. dsRNA was incubated with 5-fold dilutions of recombinant maltose-binding protein (MBP)-Loqs or MBP-GFP and complexes were resolved on native polyacrylamide gels. **B.** Quantification of (A) with dissociation constants for the indicated protein. **C.** Electrophoretic mobility assay of *Ae. aegypti* Loqs-PA (left panels) and Loqs-PB (right panels) with the indicated ssRNAs corresponding to DENV2 5' UTR, NS1, NS5 and 3' UTR sequences. Top panel indicates a schematic representation of the position of the probes on the DENV genome (not to scale). ssRNA was incubated with 2-fold dilutions of recombinant MBP-Loqs and complexes were resolved on native polyacrylamide gels. The images of the free ssRNA and the Loqs-RNA complex for the 5' UTR are cropped from the same gel. Full images are provided in S5 Fig. **D.** Quantification of (C) with dissociation constants for the indicated RNAs.

weakly interact with DENV NS1 [10], suggesting that the interaction may play a role in both mammals and mosquitoes.

While our manuscript was under revision, Shivaprasad *et al.* reported findings in support of our conclusions [41]. In that study, Loqs was identified using an unbiased approach to identify cellular proteins interacting with the DENV 3'UTR in mosquito cells. Like our results, Loqs was found to be a proviral host factor in *Aedes* cells, affecting viral replication but not

RNA stability [41]. Interestingly, Loqs was confirmed as a proviral factor for *Aedes*-associated DENV, ZIKV, and yellow fever virus, but not for the *Culex*-associated West Nile virus, suggesting specificity for *Aedes*-associated flaviviruses.

Like in *Drosophila* [25–27], there seems to be functional specialization among *Loqs* paralogs in *Aedes* mosquitoes. Loqs-PA was proposed to be essential for miRNA function, and R2D2 and Loqs-PB to be involved in siRNA production, although the relative importance of the isoforms could not be fully resolved [24]. Later, it was found that *Aedes* encodes another paralog, *Loqs2*, the product of which interacts with both R2D2 and Loqs, and is essential for antiviral defense [23]. Interestingly, *Loqs2* has a more restricted expression than *R2D2* and *Loqs* and it is not expressed in the midgut, the site of initial replication of arboviruses in the mosquito, whereas other RNAi pathway genes, including *Loqs* and *R2D2* are. This suggests that complex dynamics between proviral Loqs and antiviral Loqs2 may take place in mosquitoes and it would require further investigation to determine how Loqs affects viral replication *in vivo*.

Three *Loqs* isoforms are annotated in the current *Ae. aegypti* reference genome (AaegL5), of which the isoform with only two dsRBDs, *Loqs-RC*, is only lowly expressed and did not affect DENV replication in our experiments. We recovered one peptide specific to isoform Loqs-PA in the NS1 interactome and *Loqs-RA* silencing reduced DENV replication. However, we expect Loqs-PB to be proviral as well, as it bound DENV RNA and localized to replication organelles to a similar extent as Loqs-PA.

As expected, Loqs interacted efficiently with dsRNA, but we found it to also bind with high affinity to ssRNA corresponding to the DENV 3' UTR. The 3' UTR is extensively structured in diverse stem-loop structures, including pseudoknots (PK1 and PK2), dumbbell-like structures (DB1 and DB2), as well as a long 3′ terminal stem-loop (3′ SL) [42,43]. The 3′ SL mediates RNA-RNA interactions between the 5' and 3' ends of the genome, essential for viral RNA replication. Specifically, interaction of a cyclization sequence upstream of the initiator AUG, called the upstream AUG region (5′ UAR), with a complementary sequence in the stem of the 3' SL (3' UAR) mediates genome circularization required for initiation of negative-strand RNA synthesis [44,45]. It is likely that interaction of Loqs with the 3' UTR affects accessibility and stability of RNA structures or the transition between linear and circularized states of the DENV genome akin to eukaryotic translation elongation factor (eIF1A) that interacts with the 3'SL to mediate flavivirus negative-strand RNA synthesis in mammals [46].

NS1 interacts with other viral non-structural proteins, is essential for early events in viral RNA replication and replication organelle formation, and colocalizes with viral dsRNA [12,47,48]. We identified Loqs as an interactor of NS1, expressed as part of a polyprotein with other non-structural proteins flanked by non-viral UTRs, suggesting that Loqs was recruited to NS1 via protein-protein interactions. In contrast, Loqs was not identified in the interactome of NS1 expressed individually in mosquito cells [33]. Given that NS1 is an ER resident protein secreted through the secretory pathway [49,50] and that Loqs is a cytoplasmic protein, the interaction of Loqs with NS1 is likely to be an indirect one, mediated by other viral non-structural proteins interacting with NS1. Indeed, Shivaprasad *et al.* detected NS3 by western blot analyses of immunoprecipitations of HA-tagged Loqs (NS1 was not tested) [41] and we detected NS1, NS3 and NS5 in immunoprecipitations of GFP-tagged Loqs by mass spectrometry.

In addition to Loqs, we found multiple other interactors of NS1 and NS5 that merit further investigation, including the putative pro- and antiviral factors identified in the functional screen. Proviral E3 ubiquitin-protein ligase HYD (UBR5) was recently identified as a proviral host factor of ZIKV in a capsid interactome study in mosquitoes [51], and the human ortholog UBR5 was also found to interact with DENV NS5 [13], suggesting HYD/UBR5 to be a broad cofactor for flaviviruses. The independent pull-down of HYD with the capsid and NS5 proteins

in, respectively, the Gestuveo *et al*. study [51] and the present study suggests a prominent role of ubiquitin metabolism in flavivirus replication in mosquitoes. The identification of several proteins associated with ubiquitin metabolism (proteasome subunits β1, β2, α5 and α6, piccolo, mahjong and proviral muskelin) in our NS5 immunoprecipitations agrees with previous reports suggesting that NS5 functions as an adaptor for the ubiquitination system in humans, as it interacts with several E3-ligases, including UBR4 to promote the degradation of STAT2 [13,52].

Among the potential antiviral factors from our screen are the RNA binding proteins DIP1, an ortholog of human ADAR proteins to which pro- and antiviral activities have been attributed [53,54], and RPL12 (RPA12), which could modulate viral replication through its dsRNA cleavage and transcriptional termination activities [55]. In addition, we found the ER retention protein RER1, involved in protein degradation via the proteasome [56] and the chaperone protein HSP60A [57] to be antiviral factors, possibly related to a role of the stress response in antiviral immunity [58]. Yet, these observations need further investigation, especially since chaperone proteins, including HSP60, were proposed as flavivirus proviral cofactors in mammals [59,60]. Moreover, we found a subunit of the OST complex, OstΔ (RPN2), to be antiviral activity in mosquito cells, which is in contrast with the proviral activity of the OST complex in mammals [10,55,61]. The OST complex has previously been associated with the immune response [62,63] and could represent a host factor with different activities in vertebrate and invertebrate hosts.

Viruses recruit host factors for viral replication and several dsRNA binding proteins have been shown to interact with the 3' UTR of flaviviruses [64,65]. Despite its role as a regulator of RNAi pathways central to antiviral immunity, our work and that of Shivaprasad *et al*. [41] establishes that the RNA binding protein Loqs is hijacked through protein-protein and protein-RNA interactions to promote flavivirus replication in mosquitoes. Our study provides novel insights into the mechanisms of replication of pathogenic flaviviruses and identifies Loqs as a potential target to develop strategies to block flavivirus transmission by vector mosquitoes.

## Methods

### Plasmids

DENV2 (strain NGC) non-structural genes were PCR amplified in three fragments from the pRepDVRluc plasmid [39] and all fragments were inserted into the pPUb-MCS-2A-Puro plasmid [66] using In-Fusion cloning kit (Takara) to generate the pPUb-NS(Ø) plasmid. pPUb-NS(Ø) was subsequently mutated by site-directed mutagenesis to insert a cassette containing AgeI and SphI restriction sites. A 3xFLAG tag was then inserted by In-Fusion ligation of annealed oligos either at the N-terminus of NS1 to generate plasmid pPUb-NS(1F) or at the C-terminus of NS5 to generate pPUb-NS(5F). A pPUb-GFP plasmid was generated by removing the Gateway cassette in the pUGW vector [66], followed by the introduction of restriction sites for SphI, AvrII, AflII and NheI included in primers. *Ae. aegypti Loqs-RA* and *Loqs-RB* sequences (AaegL5) were cloned from the plasmids pPUb-HA-Loqs-PB and pPUb-HA-Loqs-PA (AaeL3), kindly provided by Zach Adelman (Texas A&M University) [24]. Loqs isoforms were inserted using the In-Fusion cloning kit into the pPUb-GFP plasmid to express recombinant proteins with a GFP tag at the C-terminus and generate plasmids pPUb-Loqs-PA-GFP and pPUb-Loqs-PB-GFP, or into the pMal-C2X (NEB) to express recombinant proteins with a maltose-binding protein (MBP) tag at the N-terminus and generate pMal-Loqs-PA and pMal-Loqs-PB. GFP-tagged Loqs-PA was further cloned into the pPUb-MCS-2A-Puro plasmid to generate pPUb-LoqsPA:GFP-2A-Puro.

The pAc-Cas9-AalbU6.2 plasmid was generated by replacing the *D. melanogaster* U6 promoter in pAc-sgRNA-Cas9 (provided by Ji-Long Liu, Addgene plasmid #49330) [67] with the *Ae. albopictus* U6.2 promoter of AALFPA_045636 and by introducing XbaI restriction sites using In-Fusion cloning (Takara) on four fragments amplified by the In-Fusion primers in S3 Table, according to the manufacturer's instructions. Single guide (sgRNA) sequences targeting *Loqs* were cloned directly after the U6 promoter into the SapI restriction site, followed by ligation of annealed and phosphorylated complementary oligos (S3 Table) using T4 DNA ligase (Roche).

All primers used for cloning are listed in S3 Table.

## Cells and viruses

*Aedes aegypti* Aag2 (kindly provided by Raul Andino, University of California San Francisco) were cleared of persistent CFAV infection to generate the Aag2-C3 clone [68] and further cleared of persistent PCLV following the same method to generate Aag2-C3PC12 cells. *Aedes albopictus* C6/36 cells (ECACC General Cell Collection, #89051705) and U4.4 cells (kindly provided by Gorben Pijlman, Wageningen University, the Netherlands) cells were maintained at 28 ˚C in Leibovitz L15 medium (Gibco) supplemented with 10% heat inactivated fetal calf serum (FCS, Sigma), 2% tryptose phosphate broth solution (Sigma), 1x MEM non-essential amino acids (Gibco), and 50 U/ml penicillin and 50 μg/ml streptomycin (Gibco). Aag2-C3PC12 cells were transfected with pPUb-LoqsPA:GFP-2A-Puro and treated with 4 μg/mL puromycin (InvivoGen) to generate a stable cell line expressing GFP-tagged Loqs-PA.

DENV serotype 2 (DENV2, strain 16681) was provided by Beate Kümmerer (University of Bonn). ZIKV (strain H/PF/2013) was obtained from the European Virus Archive (EVAg, catalog number 001v-EVA1545). Unless specified otherwise, all viral infections were performed at an MOI of 0.1 in L15 medium without FCS and the medium was refreshed at 1 h post-infection with supplemented medium, containing 2% FCS. DENV and ZIKV stocks were prepared on C6/36 cells and titrated by end-point dilution on BHK-15 or Vero E6 cells, respectively, as described previously [39].

## Immunoprecipitation

For standard immunoprecipitation of NS1 and NS5, C6/36 cells were seeded in a 60 mm culture dish and incubated for 24 h. Cells were then transfected with 30 μg of pPUb-NS(Ø), pPUb-NS(1F) or pPUb-NS(5F) and 20 μl of X-tremeGENE HP (Roche) and medium was refreshed after 3 hours. At 24 h after transfection, cells were harvested and washed in ice-cold PBS using 5 min centrifugations at 900 x *g*. Cell pellets were resuspended in 400 μL FLAG lysis buffer (10 mM Tris-HCl pH 7.4, 150 mM NaCl, 0.5 mM EDTA, 0.5% NP-40, 1x protease inhibitor cOmplete (Roche)). Samples were incubated with end-over-end rotation for 1 h at 4 ˚C and followed by centrifugation at 18,000 x *g* for 30 min at 4 ˚C. Total protein extracts were collected, diluted with 600 μL of dilution buffer (FLAG lysis buffer without NP-40) and incubated with 25 μL of pre-washed Anti-FLAG M2 Magnetic Beads (Sigma) on a rotor overnight at 4 ˚C. The beads were washed three times in ice-cold dilution buffer, eluted in 50 μL of 2X SDS containing 5% β-mercaptoethanol at 95 ˚C for 10 min and analyzed by western blot.

For mass spectrometry analysis of FLAG-tagged NS1 and NS5 interactomes compared to untagged proteins, immunoprecipitation was performed as described above, with the following modifications. C6/36 cells were seeded in 150 mm culture dishes (four dishes per condition) and transfected with 30 μg of plasmids pPUb-NS(Ø), pPUb-NS(1F) or pPUb-NS(5F) and 60 μl of X-tremeGENE HP per dish. Cells were harvested at 24 h after transfection and washed in 5 mL of ice-cold PBS and cell pellets were resuspended in 2 mL of lysis buffer. Total protein extracts were diluted with 3 mL of dilution buffer and protein concentrations were estimated

using the BCA Protein Assay (Pierce). 6 mg of total protein extract was incubated with 40 μL of Anti-FLAG M2 Agarose Beads (Sigma) and beads were washed a total of five times (three times in ice-cold dilution buffer and twice in ice-cold PBS), followed by on-bead trypsin digestion prior to MS analysis, as previously described [69]. For each condition, three biological replicates were used.

For the Loqs-PA mass spectrometry analysis, immunoprecipitation was performed as described above, with the following modifications. Aag2 stably expressing GFP-tagged Loqs-PA were seeded in T75 flasks and infected with DENV at an MOI of 0.1 for 96 h. Cells were resuspended in 1 ml GFP lysis buffer (50 mM Tris-HCl pH 7.4, 150 mM NaCl, 0.5 mM DTT, 1% NP-40, 10% glycerol and 1x protease inhibitor cOmplete (Roche)). Immunoprecipitation using GFP-Trap or binding control magnetic agarose beads (Chromotek) was performed according to manufacturer's instructions.

## Western blotting

Protein samples were separated at 120 V in 7.5% polyacrylamide gels for 90 min and transferred at 80 mA to PVDF membranes overnight at 4 ˚C using the Bio-Rad wet blotting system. Proteins were labeled with primary antibodies mouse anti-FLAG M2 (Sigma, F1804) and rat anti-tubulin-α (MCA78G) at 1:1000 dilutions, and secondary antibodies IRDye 800 goat anti-rat (Li-Cor, 926–32219) and IRDye 680RDye goat anti-mouse (Li-Cor, 926–32220) at 1:10,000 dilutions. Western blots were imaged on the Odyssey CLx System (LI-COR).

## Sample preparation and mass spectrometry

Samples were subjected to on-bead digestion [70], as follows: 50 μl of Elution buffer (EB: 2M urea, 100 mM Tris-HCl pH 8.0 and 10 mM DTT) was added to each sample and incubated for 20 min at room temperature. Cysteines were alkylated using 50 mM iodoacetamide for 10 min, after which 0.25 μg of MS-grade trypsin (Promega) was added per sample. After 2 h incubation in a thermoshaker at room temperature, the supernatants were collected. Beads were washed once with 50 μl EB to collect as many peptides as possible, and this supernatant was combined with the first, after which 0.1 μg trypsin was added and samples were incubated overnight. The next day, samples were subjected to STAGE-tipping [71]. Discs of C18 material were fixated in p200 pipette tips. The C18 material was activated using methanol, and then thoroughly washed once with buffer B (80% acetonitrile, 0.1% TFA) and twice with buffer A (0.1% TFA), after which the samples were loaded. Salts from the digestion buffer were washed away by an additional buffer A wash. Samples were eluted using buffer B for measurements.

FLAG- and GFP-immunoprecipitated samples were analyzed using reverse phase chromatography on an EASY-nLC1000 instrument coupled online to a Thermo Exploris 480 mass spectrometer. A 60 min gradient of buffer B (80% acetonitrile, 0.1% TFA) was applied to gradually release peptides from the C18 column into the mass spectrometer, which was ran at Top20 mode. A dynamic exclusion list was enabled for 30 proteins for 45 seconds after first occurrence. Only peptide ions with a charge between 2 and 6 were selected for fragmentation.

## Mass spectrometry data analysis

The raw mass spectrometry data were analyzed using MaxQuant version 1.6.0.1 [72]. FLAG-immunoprecipitated data were compared to a database for *Ae. albopictus* (Aalbo_primary.1, RefSeq assembly: GCF_006496715.1) and DENV 16681 proteins. GFP-immunoprecipitated data were compared to a database for *Ae. aegypti* (AaegL5.3 proteome downloaded from VectorBase, dataset DS_cc8d875d2e) and DENV 16681 proteins. In addition to default settings, Deamidation (NQ) was used as a variable modification, and LFQ and iBAQ were enabled.

Perseus [73] was used for filtering. Contaminants, reverse hits and hits with less than one peptide were removed. LFQ-values were subsequently log2-transformed, and samples were divided into triplicates and filtered to have at least three valid values in one group of replicates. The missing data were imputed using default settings. Students *t*-tests were performed for each of the baits compared to the control. R was used to visualize the data in volcano plots and a heatmap. When available, names of the closest ortholog from *D. melanogaster* as indicated in VectorBase were used to refer to mosquito proteins.

The mass spectrometry proteomics data have been deposited to the ProteomeXchange Consortium via the PRIDE partner repository [74] with the dataset identifiers PXD031112 and PXD035419. In the PRIDE accession PXD031112, samples are referred to as "control", "1_flagip" and "5_flagip" for NS(Ø), NS(1F) and NS(5F), respectively. In PRIDE accession PXD035419, samples are referred to as "control" and "GFP" for non-specific beads and anti-GFP beads, respectively.

### Immunofluorescence assay

C6/36 cells were seeded on coverslips in 24-well plates 24 h before introducing transgenes. Cells were transfected with 1 μg of pPUb-NS(Ø), pPUb-NS(1F) or pPUb-NS(5F) using 1 μL of X-tremeGENE HP (Roche) and incubated for 3 h before refreshing the medium. For infection experiments, Aag2 cells were infected 24 h after plating with DENV2 or mock for 72 h before transfection with pPUb-Loqs-PA-GFP, pPUb-Loqs-PB-GFP. At 24 h after transfection, cells were washed with PBS and fixed in 2% paraformaldehyde for 15 min at 4˚C. Samples were further treated at room temperature and washed in PBS containing 0.05% Tween-20 between each of the following step. Cells were permeabilized with 0.1% Triton-X100 in PBS and treated with blocking buffer (2% normal goat serum, 0.1% Triton-X100, 100 mM glycine in PBS) for 30 min. Samples were stained with primary mouse antibodies anti-FLAG M2 (Sigma, F1804) or anti-dsRNA J2 (Jena Bioscience, RNT-SCI-10010500), followed by secondary goat anti-mouse antibodies conjugated with Alexa Fluor 488 (ThermoFisher, A11001) or Alexa Fluor 594 (ThermoFisher, A11005). All antibodies were diluted 1:200 in blocking buffer. Nuclei were stained with Hoechst (5 μg/ml) for 15 min. Slides were mounted in Mowiol and stored at 4 ˚C before imaging. Confocal images were acquired using a Zeiss LSM900 microscope and analyzed with Icy ICY Imaging [75]. Uncropped images are shown in S6 Fig.

### Bioinformatic analysis

Interaction networks for proteins of interest were predicted with STRING v10 [76], considering a medium confidence (0.4) and a false discovery rate stringency of 5%. Networks were visualized with Cytoscape [77]. The genes of interest were analyzed separately with DAVID 6.8 [78] to identify enrichments for specific GO terms from the "biological process" (BP) or "molecular function" (MF) categories, using a *p*-value < 0.1 as cutoff and containing at least three members per group. The complete GO term analysis are presented in S2 and S4 Tables.

Sequences of Loqs isoforms and orthologs were aligned using MUSCLE [79], alignments were curated with Gblocks [80] and a maximum likelihood phylogenetic tree was built with PhyML and aLRT [81] using phylogeny.fr [82]. The phylogenetic tree was then visualized on iTOL [83]. Percentage identity were determined with Mview [84]. Reference sequences used are listed in S5 Table.

### *In vitro* transcription

The pRepDV2Rluc plasmid [39] was linearized with XbaI and used as template for *in vitro* transcription of replicon RNA using the T7 RiboMAX Large Scale RNA Production System

(Promega) in the presence of Ribo m7G Cap Analog (Promega) at a cap analog to GTP ratio of 2.5. Replicon RNA was purified with the RNeasy Mini Kit (QIAGEN).

Templates for *in vitro* transcription of gene specific 300–500 nt dsRNA and DENV2 ssRNA or dsRNA were generated by PCR, introducing a T7 promoter sequence or a universal tag at both 5' and 3' ends, or only at the 5' end for ssRNA using the GoTaq Flexi DNA Polymerase (Promega). If present, the universal tag was then used in a second PCR to add T7 promoter sequences at both ends of the amplicon. T7 PCR products were used as a template for *in vitro* transcription by T7 RNA polymerase for 4 h at 37 °C. The RNA was denatured at 95 °C for 10 min and gradually cooled to room temperature for annealing. Annealed dsRNA and EMSA probed were purified with the GenElute Mammalian Total RNA Miniprep Kit (Sigma) and quantified with the Nanodrop-1000 Spectrophotometer (ThermoFisher).

Primers used for in vitro transcription are listed in S3 Table.

## dsRNA-mediated gene silencing

Aag2 or U4.4 cells were plated in 48-well plates and transfected with 100 ng of dsRNA per well using 0.4 µL of X-tremeGENE HD (Roche). The medium was refreshed after 3 h and cells were incubated for 48 h. For infections, cells were transfected again with dsRNA as described above, incubated for 3 h, and then infected with DENV at an MOI of 0.1. Cells were harvested at the indicated time points for total RNA isolation using RNA-Solv (Omega Bio-tek). A similar setup was used for the experiments of Fig 4G, 4I and 4L, except that cells were cultured in 24-well plates, treated twice with 150 ng of dsRNA, infected at an MOI of 0.01, after which cells or culture supernatant was harvested at 72 h post-infection, as indicated. For replicon assays, Aag2 and U4.4 cells were transfected with 100 ng dsRNA and 250 ng RepDV2Rluc RNA per well using the TransIT-mRNA transfection kit (Mirus). As a positive control, cells transfected with RepDV2Rluc RNA in the absence of dsRNA were treated with 50 µM cycloheximide (CHX) at 1 h post-transfection. Proteins were harvested in passive lysis buffer (Promega) at indicated times and Renilla luciferase activity was measured using the Renilla-Glo Luciferase Assay system (Promega).

## Reverse transcription and quantitative PCR

Transcript annotation according to the reference genome AaegL5.2 was used to generate PCR primers. For RT-qPCR, 200–500 ng of total RNA were treated with DNaseI (Ambion) for 45 min at 37 °C and then incubated with 2.5 mM EDTA for 10 min at 75 °C. Total RNA was reverse transcribed with random hexamers using the Taqman reverse transcription kit (Applied Biosystems). Negative-strand specific viral cDNA was synthesized using a virus specific primer with a flanking sequence of the T7 promoter, followed by qPCR using a T7 promoter primer in combination with a virus specific primer. Relative quantitative PCR analysis was performed using the GoTaq qPCR SYBR mastermix (Promega) on a LightCycler 480 instrument (Roche). Target gene expression levels were normalized to the expression of the housekeeping gene, *lysosomal aspartic protease* (LAP) for *Ae. aegypti* or *ribosomal protein L5* (RPL5) for *Ae. albopictus*, and fold changes were calculated using the using the $2(-\Delta\Delta CT)$ method [85].

Primers used for RT-qPCR are listed in S3 Table.

## Preparation of small RNA libraries

Small RNA deep sequencing libraries were generated using the NEBNext Small RNA Library Prep Set for Illumina (E7560, New England Biolabs), using 1 µg RNA as input. Libraries were

prepared in accordance with the manufacturer's instructions and sequenced on an Illumina Hiseq4000 by the GenomEast Platform (Strasbourg, France).

## Small RNA sequence analysis

The initial quality control was performed using FastQC and 3' adapters were trimmed using cutadapt [86]. Small RNA sequences in the size range of 21–23 nt were considered as siRNA and 25–32 nt considered as piRNAs. Reads were uniquely mapped to the corresponding virus genome, DENV2 (NCBI Reference Sequence: NC_001474.2) or ZIKV (GenBank: KJ776791.2), allowing one mismatch. SAMtools was used to quantify piRNAs mapped to the tapiR1 locus. To compare the vsiRNA changes upon *Loqs* silencing, uniquely mapped siRNA reads were normalized to piRNAs mapping to the tapiR1 locus [87] to account for differences in library size, and then normalized to DENV or ZIKV RNA copies based on RT-qPCR performed on the RNA that was used for small RNA sequencing.

Small RNA sequences in the size range of 19–25 nt were mapped to *Aedes* mature miRNA and pre-miRNA sequences using Bowtie [88], allowing a maximum of 1 mismatch. miRNA sequences and accession numbers were from the miRBase repository [89]. To compare total miRNA changes upon *Loqs* knockdown, miRNA reads were normalized to piRNAs mapped to the tapiR1 locus or total small RNA library size. SAMtools were used to quantify miRNA counts per sample [90]. DEseq2 was used for differential analysis of miRNA between samples [91].

## CRISPR/Cas9

To obtain *Loqs* and *Ago2* double knockout (KO) cells, U4.4-*Ago2*^KO cells (described elsewhere, manuscript in preparation) were transfected with the pAc-Cas9-AalbU6.2 plasmid, encoding 3xFLAG-tagged Cas9 with N- and C-terminal SV40 nuclear localization signals, followed by a viral 2A self-cleavage peptide and the puromycin N-acetyltransferase, driven by the *D. melanogaster* Actin 5c promoter, as well as a sgRNA targeting *Loqs* driven by an *Ae. albopictus* U6 promoter. U4.4-*Ago2*^KO cells were seeded in a 24-well plate and transfected the next day with 500 ng of plasmid, using X-tremeGENE HP transfection reagent (Roche) according to the manufacturer's instructions. At 2 days after transfection, puromycin (InvivoGen) was added to the culture medium at a concentration of 20 μg/ml and 4 days later, the cells were transferred to a new plate at a 1:2 dilution in medium containing 20 μg/ml puromycin. The other half of the cells was used for genomic DNA isolation (Zymo Research #D3024) and PCR (Promega #M7806) using primers flanking the sequence targeted by the sgRNA (S3 Table) to assess editing efficiency. Multiple sgRNA constructs were initially generated and the constructs with the highest editing efficiency were selected, assessed by size heterogeneity of the PCR products on an ethidium bromide-stained agarose gel. Cells transfected with these sgRNA constructs (guide 1 and guide 3) were seeded in 96-well plates at a density of a single cell per well in supplemented L15 medium in the absence of puromycin. After 3 weeks, genomic DNA was isolated from the single-cell clones, followed by PCR and Sanger sequencing of the targeted *Loqs* locus. Based on the sequencing results, a clone containing only out-of-frame deletions in the *Loqs* coding sequence (guide1, clone #1) was selected for experiments, referred to as U4.4-*Ago2*^KO; *Loqs*^KO cells.

## Production and purification of recombinant protein

*E. coli* strain XL10 Gold were transformed with plasmids encoding MBP-Loqs-PA or MBP-Loqs-PB and cultured until mid-log phase ($OD_{600}$ of 0.6). Expression of recombinant proteins was then induced with 1 mM isopropylβ-D-1-thiogalactopyranoside (IPTG) and

cells were cultured overnight at 27 ˚C. Cells were pelleted (13,000 x g, 15 min) and resuspended in *E. coli* Lysis Buffer (PBS, 0.5% (w/v) Tween-20, 1 mM EDTA, 1x protease inhibitor cOmplete). Cells were subjected to three freeze (-80 ˚C)-thaw (37 ˚C) cycles and sonicated (Branson Sonifier 250, 10 seconds, 3 x 5 cycles) before clearing debris by centrifugation at 13,000 x g for 30 min. Recombinant proteins were affinity-purified using amylose resin using the manufacturer's protocol (NEB) and eluted with 20 mM maltose. Protein concentration was measured in eluate fractions by a Bradford assay (Bio-Rad) and the fractions with the highest concentration were transferred to Slide-A-Lyzer dialysis cassette (Thermo Fisher) and dialyzed to buffer (20 mM Tris-HCl pH 7.4, 0.5 mM EDTA, 5 mM MgCl2, 1 mM DTT, 140 mM NaCl, 2.7 mM KCl). Protein concentration was determined by a Bradford assay. Recombinant proteins were snap-frozen in liquid nitrogen and stored at -80 ˚C in dialysis buffer with 30% glycerol.

## Electrophoretic mobility shift assays (EMSA)

The synthesized probes were treated with DNaseI (Promega), dephosphorylated (Roche) and end-labeled using T4 polynucleotide kinase (NEB) with [$\gamma$-$^{32}$P] ATP (Perkin Elmer). Unincorporated nucleotides were removed with MicroSpin G-50 columns (Illustra). Samples were then heated for 5 min at 85 ˚C, cooled to room temperature and incubated for 20 min in RNA folding buffer (111 mM HEPES, 6.7 mM $MgCl_2$, 111 mM NaCl). Purified proteins were diluted in dialysis buffer and incubated for 30 min at room temperature with 1–10 ng of the labeled RNA in binding buffer (5 mM HEPES, 25 mM KCl, 2 mM MgCl2, 3.8% glycerol) in the presence of 0.625 mg/mL yeast tRNA (Sigma). The reactions were loaded on a 6% native acrylamide gel and run at 4 ˚C. The radioactive signal was quantified using a Storage Phosphor Screen GP (Kodak) or GE Storage Phosphor Storage Screen BAS-IP MS 2040 E (Merck) and a Typhoon FLA 7000 biomolecular imager. $RNA_{total}$ and $RNA_{free}$ were quantified using Fiji [92] to determine the fraction bound = $1 - (RNA_{free}/RNA_{total})$. Binding isotherms were fitted using specific binding with Hill slope in GraphPad Prism7.

## Statistical analysis

Graphical representation and statistical analyses were performed using GraphPad Prism7 software. Differences were tested for statistical significance using unpaired two-tailed t-tests or one-way ANOVA, as specified.

## Supporting information

**S1 Fig. DENV non-structural proteins interact with Loqs-PA. A.** Volcano plot of proteins interacting with GFP-tagged Loqs-PA in DENV2 infected Aag2 cell lysates as determined by label-free quantitative mass spectrometry. The X-axis shows the log2 fold change (FC) of anti-GFP immunoprecipitation (IP) over control IP using the same lysate and non-specific beads (control), and the Y-axis shows -log10(*p*-value). Hits with a log2(FC) higher than 7 or -log10 (*p*-value) higher than 6 were labeled. Colored dots indicate proteins of interest. Each condition was performed in triplicate. **B.** Functional STRING networks of Loqs interacting proteins (top right quadrant in (A)) based on *Ae. aegypti* annotation, using the following three sources: *experiments*, *databases* or *co-expression*. Interactions from FLAG-NS1 and FLAG-NS5 immunoprecipitations (Fig 1) were added in purple. Nodes are scaled to the fold enrichment in Loqs immunoprecipitation and colored according to the presence of major RNA-interacting domains defined by DAVID clustering of *D. melanogaster* orthologs. Edges are representative of the number of sources (thickness) and the confidence (color) supporting the interaction as

defined by STRING. Functional groups were imputed from DAVID clustering and literature analysis.
(EPS)

**S2 Fig. Expression of Loquacious isoforms in *Ae. aegypti* mosquitoes and depleted cells. A.** Density plot of *Loqs* RNA seq reads in *Ae. aegypti*. Each line represents a unique sequence library from three separate VectorBase datasets: DS_dcde6b4ec9, DS_24f2db6f66 and DS_ded344cb5e [93–95]. **B-D.** Relative quantification of targeted gene mRNA expression from Figs 4E (S1B), 4F (S1C), 4G and 4I (S1D), and 4J, 4K, S2F and S2G (S1E). Gene expression was normalized to the housekeeping gene *lysosomal aspartic protease* and expressed relative to expression in cells treated with control dsRNA targeting luciferase (GL3). **F.** Total miRNAs at 48 h after infection at an MOI of 0.1 in *Loqs* depleted or control (GL3 luciferase dsRNA treated) Aag2 cells. Total miRNA reads were normalized to tapiR1. **G.** Correlation of *Ae. aegypti* miRNA levels in GL3 (x-axis) and Loqs (y-axis) dsRNA treated Aag2 cells infected with DENV (left panel) or ZIKV (right panel). The sum of raw read counts of miRNAs across three small RNA library replicates were calculated via Samtools. Highlighted are miRNAs with > 2-fold differential expression with an adjusted p-value < 0.05 using DEseq2. RPM, reads per million.
(EPS)

**S3 Fig. Knockout of *Loqs* in U4.4-*Ago*^KO cells.** U4.4-*Ago2*^KO; *Loqs* ^KO cell lines were generated by CRISPR/Cas9-mediated editing of the *Loqs* gene in U4.4-*Ago2*^KO cells, single-cell colonies were grown, and the edited sites in exon 2 were Sanger sequenced. Top panel, gene structure and conserved domains of two *Loqs* paralogs annotated in the genome sequence of the *Ae. albopictus* C6/36 cell line [96] (AALC636_029187 and AALC636_022034), each of which predicted to encode a long and short isoform. The coding sequences of the two paralogs are 99.2% identical at the nucleotide level and both are expected to be targeted by the sgRNAs used for CRISPR/Cas9 editing. Bottom panel, sequencing identified one clone (guide 1, clone #1) containing two small deletions that both induced out-of-frame mutations in the first dsRBD of *Loqs*.
(EPS)

**S4 Fig. Purification of recombinant Loqs-PA and–PB.** Coomassie blue stained polyacrylamide gel containing lysates from the indicated purification steps of *Ae. aegypti* MBP-tagged Loqs-PA and Loqs-PB from *E. coli*.
(EPS)

**S5 Fig. Uncropped gel images. A.** Uncropped images of the western blots from Fig 1A. **B-C.** Uncropped images of the EMSAs from Fig 6A and 6C.
(TIFF)

**S6 Fig. Uncropped IFA images.** Uncropped images of IFAs from Fig 5. Only cells with low to moderate expression of Loqs-GFP were analyzed.
(EPS)

**S1 Table. NS1 and NS5 mass spectrometry hit summary.** List of proteins enriched at least 2.5-fold in NS1 or NS5 immunoprecipitations with their -log10(*p*-value) and log2(fold change) values plotted on the volcano plot (Fig 1D), the LFQ after row-mean subtraction plotted on the heatmap (Fig 1E) as well as protein and gene identifiers in *Ae. albopictus* and references to orthologs in *Ae. aegypti*, *D. melanogaster* and *H. sapiens*. Individual values plotted on the heatmap were labeled when below (blue) or above (red) the row average. Proteins associated with both NS1 and NS5 are highlighted in yellow. Hits included in the STRING network (Fig 2)

and GO terms are indicated. Ost48 and Stt3A were included after literature review.
(XLSX)

**S2 Table. GO term analysis data.** Complete GO term analysis of Fig 2A, including GO terms for biological process (BP), cellular component (CC) and molecular function (MF).
(XLSX)

**S3 Table. List of oligonucleotides used for cloning, dsRNA production, qPCR, PCR and EMSA probes.**
(XLSX)

**S4 Table. Loqs mass spectrometry hit summary.** List of proteins enriched at least 3.5-fold in Loqs immunoprecipitation with their -log10(*p*-value) and log2(fold change) values plotted on the volcano plot (S1 Fig). Protein and gene identifiers in *Ae. aegypti* and references of *D. melanogaster* orthologs were specified. ID used for STRING network analysis and DAVID clustering, and names used in S1 Fig are specified.
(XLSX)

**S5 Table. Reference sequences.** List of protein sequences used for the phylogenetic tree of Fig 4A.
(XLSX)

## Acknowledgments

We thank members of the laboratory for discussions. We thank Pascal Jansen for processing the mass spectrometry samples. We thank Zach Adelman (Texas A&M University) for kindly providing Loqs expression plasmids and Beate Kümmerer (University of Bonn) for providing DENV2 stock. ZIKV isolate H/PF/2013 was provided by Xavier de Lamballerie (Aix Marseille Université) through the European Virus Archive (EVAg), funded by the European Union's Horizon 2020 programme. The Vermeulen lab is part of the Oncode Institute, which is partly funded by the Dutch Cancer Society.

## Author Contributions

**Conceptualization:** Benoit Besson, Ronald P. van Rij.

**Formal analysis:** Benoit Besson, Jieqiong Qu, Ronald P. van Rij.

**Funding acquisition:** Benoit Besson, Ronald P. van Rij.

**Investigation:** Benoit Besson, Oscar M. Lezcano, Gijs J. Overheul, Kirsten Janssen, Cornelia G. Spruijt, Jieqiong Qu.

**Methodology:** Benoit Besson, Gijs J. Overheul, Cornelia G. Spruijt, Jieqiong Qu, Ronald P. van Rij.

**Project administration:** Benoit Besson, Ronald P. van Rij.

**Resources:** Michiel Vermeulen, Ronald P. van Rij.

**Supervision:** Benoit Besson, Michiel Vermeulen, Ronald P. van Rij.

**Visualization:** Benoit Besson, Oscar M. Lezcano, Gijs J. Overheul, Kirsten Janssen, Cornelia G. Spruijt, Jieqiong Qu.

**Writing – original draft:** Benoit Besson, Ronald P. van Rij.

**Writing – review & editing:** Benoit Besson, Jieqiong Qu, Ronald P. van Rij.

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
