## [Decision Letter · Decision Letter 0]

11 Mar 2022

Dear Dr. Van Rij,

Thank you very much for submitting your manuscript "Arbovirus-vector protein interactomics identifies Loquacious as a co-factor for dengue virus replication in Aedes mosquitoes" for consideration at PLOS Pathogens. As with all papers reviewed by the journal, your manuscript was reviewed by members of the editorial board and by several independent reviewers. In light of the reviews (below this email), we would like to invite the resubmission of a significantly-revised version that takes into account the reviewers' comments.

We cannot make any decision about publication until we have seen the revised manuscript and your response to the reviewers' comments. Your revised manuscript is also likely to be sent to reviewers for further evaluation.

Sincerely,

Alain Kohl

Guest Editor

PLOS Pathogens

Sonja Best

Section Editor

PLOS Pathogens

Kasturi Haldar

Editor-in-Chief

PLOS Pathogens

orcid.org/0000-0001-5065-158X

Michael Malim

Editor-in-Chief

PLOS Pathogens

orcid.org/0000-0002-7699-2064

Reviewer's Responses to Questions

**Part I - Summary**

Reviewer #1: This is a very nice manuscript where the authors perform a DENV NS1-NS5 protein interactome stuydy in mosquito cells. The manuscript is well written and the figures are nicely assembled and easy to follow. Their findings demonstrate a role of Loqs during flavivirus infection of mosquito cells by 1) showing, most likely indirect, protein interations between Loqs and NS1 2) co-localization during infection 3) silencing of Loqs is detrimental to flavivirus infection 4) Loqs binds to the 3'UTR, and 5) the interactions function independently of RNAi. I have two specific comments below.

Reviewer #2: In this manuscript, Besson and colleagues systematically profile the cellular interactomes of NS1 and NS5 - two important flaviviral proteins with functions in RNA replication, replication complex formation and viral RNA modification(s) – within C6/36 cells, an RNAi-incompetent Ae. albopictus cell line. Exploiting an ectopic expression system to drive expression of the entire sub-genomic DENV viral genome (NS1-NS5) under the control of the invertebrate polyUbiquitin promoter, they generate epitope-tagged variants of the viral polyprotein, carrying NS1- and NS5-FLAG-tagged variants for use in AP-LC-MS/MS experiments. Among the specific NS1 and NS5 interactors, 22 prioritized host factors were followed-up in silencing experiments to assess their functional relevance for viral replication, and one of them – the host protein Loquacious - was investigated further by immunofluorescence, isoform-mapping and dsRNA-binding experiments. Altogether, the authors propose that LoqRA interacts with the 3’UTR of DENV genome, via recruitment through the NS1 protein and plays an important pro-viral function independent from the RNAi pathway.

The study is well-written and timely, as it addresses understudied aspects of virus-host interactions in invertebrates. However, the wealth of publications on Loquacious role in flavivirus replication, as well as other studies on NS1 and NS5-host interaction networks in mosquitoes (cited by the authors in the manuscript) partially hamper the novelty of the study. Most importantly the main conclusions and novel findings (Loquacious plays a proviral role, independent of its role in RNAi, to support replication organelle formation) are only partially supported by experimental evidence as some experiments lack important controls (i.e. EMSA) or might be interpreted differently (i.e. Luciferase assays) as outlined below.

Reviewer #3: The authors of this study report an interesting novel, and rather unexpected, pro-viral function for Loquacious (Loqs) in dengue virus (DENV) replication in cells derived from two important mosquito vector species, Aedes aegypti and Aedes albopictus. The findings are significant because they fundamentally change our view of the canonically antiviral functions of proteins functioning within the RNA interference (RNAi) pathway in insects. The work also advances our understanding of how molecular virus-vector interactions influence arthropod-borne virus (arbovirus) replication in mosquito cells, which is still a poorly developed area of research that has important implication for how significant arboviral pathogens are transmitted. The study is thorough and well executed, and uses a wide range of complementary approaches, including proteomics/interactomics and functional RNAi screens to characterise virus-vector interactions, which are then dissected in molecular detail using viruses, replicons and biochemical assays of RNA-protein binding. The manuscript is well written and the figures well presented and I believe the work makes a valuable and important contribution to the field.

**Part II – Major Issues: Key Experiments Required for Acceptance**

Reviewer #1: 1. The authors demonstrate a reduction in viral RNA, but does this translate to reduced infectious particles? It would have been nice to see plaque assay or focus forming unit data. Are samples still available that such assays could be easily performed?

2. While the authors did a lot of very nice work, I was left wondering if their cell culture observations were transferrable to mosquitoes. The impactfulness of this work would be significantly increased if the authors were able to observe decreased viral loads in mosquitoes upon Loqs suppression. However, the results might be a mixed bag because suppression of Loqs would also prosumably retard RNAi and, therefore, no differences might be observed. In leiu of performing mosquito experiments this might be an area the authors could speculate upon in the discussion section.

Reviewer #2: - It would be important to include control pull-downs using baits with a similar subcellular localization (i.e. an unrelated FLAG-tagged protein) rather than the untagged version of the polyprotein). Indeed, the specific distribution of the interactors in the volcano plots and the outcome of the statistical analysis directly reflects the use of “empty” pull-down as control. As a result, nearly all the viral proteins appear as specific interactors in both NS1 and NS5 dataset (Fig1), and both proteins share >30% of common host protein interactors (Fig2a) despite the rather divergent subcellular localization (NS1=ER vs. NS5=Cytoplasm+Nucleus). Along these lines, validation of the NS1-Loqs interaction upon infection (reciprocal IP using the available GFP-tagged Logs) would serve as important validation of the MS data.

- The readout of the silencing experiments and all downstream k.d. experiments exclusively relies on qRT-PCR on viral RNA, however it would be paramount to functionally validate at least the main host factors using an infectivity-based readout (i.e. TCID50, FFU or PFU assays). This is especially important as some experiments suggest a lack of Logs k.d. effect on virus replication (i.e. Fig4G, see comments below). Furthermore, standard deviations appear extremely high (i.e. Fig. 4E-H) and lack uninfected/NT-silenced controls.

- The conclusion that Loqs proviral activity is independent of its RNAi regulatory function (result section; from row 209 onwards) is only indirectly supported by the data. It would be straightforward to compare the outcome of Logs silencing on virus replication in both RNAi-competent (U4.4) and RNAi-deficient cells (C6/36) already available and used by the authors in this manuscript.

- The conclusion that Logs silencing does not affect RNA translation while inhibiting viral RNA replication is not experimentally supported. Indeed, experiments using the luciferase reporter virus do not show any reduction in luciferase activity upon Logs silencing, suggesting that both RNA replication and translation are not affected. If any, in U4.4 cells, silencing of Logs slightly enhances viral RNA replication (Fig 4G). Furthermore, this conclusion is not compatible with the authors’ hypothesis suggesting a role for Logs in viral RNA stability through binding of the 3’-UTR (one would expect to see a reduction here) and seemingly in contrast with qRT-PCR results presented in Fig. 3.

- The RNA binding experiments (EMSA; Fig5A) would require an additional MAP-tagged protein as control to ensure specificity and accuracy of binding affinity measurements. Alternatively, the use of a non-flaviviral RNA might suffice. These aspects are particularly important as the input RNA of all the other control probes (5’UTR, NS5, NS1; see “free XX region”), as well as higher MW bands appearing at higher concentrations of purified protein (>200nM), appear significantly lower when compared to the corresponding 3’UTR.

- Result section – line 250: the hypothesis that Loqs is recruited to the replication organelles and binds viral RNA through its dsRNA-binding motif would require experimental confirmation. Indeed, in EMSA experiments only ssRNA has been used. (see comment above).

Reviewer #3: I’m not fully convinced of the interpretation that Loqs affects viral genome replication but not viral RNA stability or translation. The data show that Loqs knockdown has no effect on transiently transfected DENV replicon, but does reduce viral RNA levels during infection. I agree that that the replicon data indicate that Loqs does not seem to affect viral RNA stability or translation, however, unless I’m mistaken this particular replicon also replicates its RNA, and therefore the replicon assay also indicates that Loqs does not affect viral RNA replication. The major difference between viral infection and the replicon assay is that the replicon bypasses viral entry and uncoating steps, and therefore it seems to me that Loqs rather affects these early stages of the viral life cycle rather than viral genome replication. I think the authors would need to perform additional experiments to support their current conclusions. For example, they could transfect the infectious viral RNA into cells and assess whether viral RNA levels are still reduced in the absence of Loqs when entry/uncoating is bypassed. Alternatively, one could measure the accumulation of negative-sense viral RNA in both the replicon and infection experiment to directly measure viral genome replication.

The authors also conclude that the pro-viral role of Loqs is independent of Loqs’s function in the RNAi pathway. The data shown does indeed indicate that Loqs knockdown does not affect the accumulation of virus-derived siRNAs. However, I do not think this is sufficient to support the claim that Loqs acts independently of the RNAi pathway. Evidence that might better support this claim would be to test the effect of Loqs knockdown in the presence of concurrent Dicer2 or Ago2 knockdown, or in Dicer2 or Ago2 CRISPR knockout cell lines (in which the RNAi pathway would not be functional). Another question I have is whether the authors saw differences in piRNA production in this experiment?

In Fig 5, the dsRNA staining shows a lot of background in mock cells – which I suspect is due to the presence of other insect viruses persistently infecting these cells. Since the authors are investigating whether Loqs localises to DENV replication compartments, could they perform a sequence-specific stain for DENV RNA to specifically stain DENV replication compartments and reduce the background noise in the mock?

**Part III – Minor Issues: Editorial and Data Presentation Modifications**

Reviewer #1: (No Response)

Reviewer #2: - Why use the C6/36 cells (RNAi-incompetent) as model system for the AP-LC-MS/MS study? The follow-up functional screen was then done in the RNAi-competent U4.4 cell line – why not directly use the U4.4 cell line?

- In the method section, row 368: “cells were transfected with 10 mg”: probably meant 10 µg ?

- Supplementary figure S2 (Protein purification): the SDS-PAGE does not show a single band for the purified protein post dialysis, but rather a smear of 20-30 kDa for both Loqs-PA and PB. Could this reflect incomplete cleavage of the MAP-tag? This might affect interpretation of the EMSA experiments.

- Supplementary Figure S4: oversaturated signals in IF images for both GFP-tagged proteins and dsRNA.

- FIG 5: there is quite significant background for dsRNA in the mock sample. Furthermore, no “empty”-eGFP control was used in this panel.

Reviewer #3: 1. In Fig 5, I found it quite difficult to see the visual differences that the authors describe in the localisation of Loqs, particularly any differences between the two isoforms. Could the authors provide some quantification here to strengthen these findings?

2. In the Materials and Methods, the authors should specify what (if any) corrections for multiple comparisons were applied in their statistical analyses.

PLOS authors have the option to publish the peer review history of their article (what does this mean?). If published, this will include your full peer review and any attached files.

Reviewer #1: **Yes: **Doug Brackney

Reviewer #2: No

Reviewer #3: No
---

## [Decision Letter · Decision Letter 1]

14 Jul 2022

Dear Dr. Van Rij,

Thank you very much for submitting your manuscript "Arbovirus-vector protein interactomics identifies Loquacious as a co-factor for dengue virus replication in Aedes mosquitoes" for consideration at PLOS Pathogens. As with all papers reviewed by the journal, your manuscript was reviewed by members of the editorial board and by several independent reviewers. The reviewers appreciated the attention to an important topic. Based on the reviews, we are likely to accept this manuscript for publication, providing that you modify the manuscript according to the review recommendations.

We will not ask for additional experiments at this stage, but would ask that any comments about interpretation or missing data/figures are addressed at revision.

Sincerely,

Alain Kohl

Associate Editor

PLOS Pathogens

Sonja Best

Section Editor

PLOS Pathogens

Kasturi Haldar

Editor-in-Chief

PLOS Pathogens

orcid.org/0000-0001-5065-158X

Michael Malim

Editor-in-Chief

PLOS Pathogens

orcid.org/0000-0002-7699-2064

We will not ask for additional experiments at this stage, but would ask that any comments about interpretation or missing data/figures are addressed at revision.

Reviewer Comments (if any, and for reference):

Reviewer's Responses to Questions

**Part I - Summary**

Reviewer #1: The authors did a very thorough job addressing the reviewers concerns through the addition of more explanatory text, extra experiments and additional controls.

Reviewer #2: The manuscript is now substantially improved, and the authors put quite some efforts in providing important missing controls on EMSA assays and modulation of piRNA. Furthermore, additional clarifications on experimental set-up s and statistical analysis were added, and some statements on interpretation of experiments were tuned down. Nonetheless, this reviewer feels that the claim that Loq plays a role in vRNA replication is still not experimentally supported (see comment below).

Reviewer #3: The authors have put considerable effort into addressing the reviewers’ comments with a number of rewrites, additional experiments and analyses. My original comments have been addressed in full and I have no further comments.

**Part II – Major Issues: Key Experiments Required for Acceptance**

Reviewer #1: None noted

Reviewer #2: This reviewer is still particularly concerned on the interpretation of the Loq k.d. results on viral RNA replication. Indeed, despite the clarifications provided by the authors, in the revised manuscript there is still no experimental evidence that Loqs plays a role in viral RNA replication.

The justification that in the authors' hands the subgenomic replicon used is “not stable/degraded” over time seems not to be experimentally supported, as data presented are normalized by the 3h R.L.U. value, which represent the input (i.e. translation of in vitro synthesized RNA). Clearly, both in Figure R1 and in Figure 4h, this replicon replicates above input values for at least 24-40h. To my knowledge a viral RNA which does not replicate should decline over time. In vertebrate cells this usually happens at 12h.p.t..

Importantly, the new sentences added in the text do not resolve this issue, but simply state that according to the authors this replicon does not replicate:

Lines 221-223: “To this end, Renilla luciferase expression was monitored after transfection of subgenomic replicon RNA into Ae. aegypti Aag2 cells, under experimental conditions in which replicon RNA is translated but not replicated (Fig. 4H)”. If such claim has to be made, comparing a R.L.U. activity of a replication-deficient replicon (such as an NS5 “GND-mutant” which by definition does not replicate but can be translated) with a replication-competent would be required.

Additionally, also the new experiment provided upon suggestion by Reviewer 3 (Fig4i), does not corroborate the hypothesis that Loq affect viral RNA replication. Indeed, the crucial experiment proposed by reviewer 3 was to “ measure the accumulation of negative-sense viral RNA in both the replicon and infection experiment to directly measure viral genome replication.”

The experiments provided in new Fig. 4i were performed only with full length viruses in conditions where multiple cycles of replication take place (MOI=0.01, readout by qRT-PCR at 48h.p.i.), thereby confounding/additive effects of viral entry, RNA replication and viral spread on the observed reduction in (-)vRNA cannot be ruled out.

This experiment, or any other unequivocally supporting the hypothesis that vRNA replication is affected is required. Alternatively, the author should rephrase the corresponding paragraphs referring to a more general effect on “viral replication”.

Reviewer #3: N/A

**Part III – Minor Issues: Editorial and Data Presentation Modifications**

Reviewer #1: None noted

Reviewer #2: - The new figure cited in the rebuttal and the main test „Fig S1d” is missing from the supplementary file

- The GFP empty control in Fig5 was required to control for bleedthrough of the 488 channel in the 568 channel rather than the colocalization. In general, resolution of Fig 5 (including the high-res downloadable file) is still very poor.

Reviewer #3: N/A

PLOS authors have the option to publish the peer review history of their article (what does this mean?). If published, this will include your full peer review and any attached files.

Reviewer #1: No

Reviewer #2: **Yes: **Pietro Scaturro

Reviewer #3: No

Figure Files:

Data Requirements:

Reproducibility:

References:

---

## [Editor Report · Decision Letter 2]

26 Jul 2022

Dear Dr. Van Rij,

We are pleased to inform you that your manuscript 'Arbovirus-vector protein interactomics identifies Loquacious as a co-factor for dengue virus replication in Aedes mosquitoes' has been provisionally accepted for publication in PLOS Pathogens.

Best regards,

Alain Kohl

Associate Editor

PLOS Pathogens

Sonja Best

Section Editor

PLOS Pathogens

Kasturi Haldar

Editor-in-Chief

PLOS Pathogens

orcid.org/0000-0001-5065-158X

Michael Malim

Editor-in-Chief

PLOS Pathogens

orcid.org/0000-0002-7699-2064
---

## [Editor Report · Acceptance letter]

1 Sep 2022

Dear Dr. van Rij,

We are delighted to inform you that your manuscript, "Arbovirus-vector protein interactomics identifies Loquacious as a co-factor for dengue virus replication in Aedes mosquitoes," has been formally accepted for publication in PLOS Pathogens.

Best regards,

Kasturi Haldar

Editor-in-Chief

PLOS Pathogens

orcid.org/0000-0001-5065-158X

Michael Malim

Editor-in-Chief

PLOS Pathogens

orcid.org/0000-0002-7699-2064